# Ecological mechanisms and current systems shape the modular structure of the global oceans' prokaryotic seascape

**Felix Milke** [1] ✉, **Jens Meyerjürgens** [1] **& Meinhard Simon** [1,2] ✉

Major biogeographic features of the microbial seascape in the oceans have been established and their underlying ecological mechanisms in the (sub) tropical oceans and the Pacific Ocean identified. However, we still lack a unifying understanding of how prokaryotic communities and biogeographic patterns are affected by large-scale current systems in distinct ocean basins and how they are globally shaped in line with ecological mechanisms. Here we show that prokaryotic communities in the epipelagic Pacific and Atlantic Ocean, in the southern Indian Ocean, and the Mediterranean Sea are composed of modules of co-occurring taxa with similar environmental preferences. The relative partitioning of these modules varies along latitudinal and longitudinal gradients and are related to different hydrographic and biotic conditions. Homogeneous selection and dispersal limitation were identified as the major ecological mechanisms shaping these communities and their free-living (FL) and particle-associated (PA) fractions. Large-scale current systems govern the dispersal of prokaryotic modules leading to the highest diversity near subtropical fronts.

Our knowledge of the taxonomic and functional biogeographic patterns of oceanic microbial communities has greatly improved in the recent past due to analyses based on expeditions covering major regions of the global oceans[1–4]. As these expeditions did not systematically study the basins of the Atlantic and Pacific Ocean, encompassing together 80% of the global ocean surface, the data collected by these expeditions do not allow a systematic assessment and comparison of the microbial biogeography in these ocean basins. For the subtropics and tropics, the similarity of prokaryotic as well as picoeukaryotic communities within ocean basins is higher than between basins[4] but it is unknown whether this can be generalized for the temperate to subpolar regions. Biogeographic patterns of pro- and eukaryotic microbial communities between subpolar and tropical regions in the Atlantic and Pacific Ocean have been assessed but both oceans have not been systematically compared[5–9]. These studies provide evidence that in temperate and frontal regions prokaryotic

richness is higher than in other regions, findings in line with a global modeling study[10].

Despite the established knowledge of the global biogeographic patterns of oceanic microbial communities, we still have a very limited understanding of how these patterns evolve. Selection, dispersal and drift are the main ecological mechanisms shaping such patterns[11,12]. Looking just at the global biogeography of the subtropical and tropical oceans based on amplicon sequence variants (ASV) of the 16 S rRNA gene it has been reported that drift is most important and homogeneous selection and dispersal limitation each contribute to half of drift to shape the structure of prokaryotic communities[3]. In the epi- and upper mesopelagic Pacific Ocean between subantarctic and subarctic regions homogeneous selection was identified as the predominant mechanism for shaping the structure of FL prokaryotic communities[9]. To understand the significance of these ecological mechanisms for shaping prokaryotic community composition in the

[1]Institute for Chemistry and Biology of the Marine Environment, University of Oldenburg, Carl von Ossietzky Str. 9-11, D-26129 Oldenburg, Germany. [2]Helmholtz Institute for Functional Marine Biodiversity at the University of Oldenburg (HIFMB), Ammerländer Heerstraße 231, D-26129 Oldenburg, Germany. ✉e-mail: felix.milke@uni-oldenburg.de; m.simon@icbm.de

global oceans and the major ocean basins, we need to assess these mechanisms in the Atlantic Ocean and other oceans in comparison to the Pacific Ocean. Further, we need to understand how they act in the context of hydrography and in particular of large-scale current systems.

The oceans are interconnected by current systems and maintain a constant redistribution of planktonic organisms on various scales. The current systems, together with ecological mechanisms and the greatly varying turnover times of microbes, mediate the global dispersal of prokaryotes and lead to the establishment of distinct microbial biogeographic patterns[1,13–15]. Physical barriers such as oceanic fronts and basin structure can reduce dispersal between adjacent water masses and favor selective processes to alter microbial communities[16–19]. Two studies linking the composition and turnover of microbial communities with surface current systems in the global oceans have been published recently based on data collected during the Tara Ocean expeditions that missed temperate and subpolar biogeographic regions both in the Atlantic and Pacific Ocean. One study, focusing on the globally most abundant cyanobacterium *Prochlorococcus*, occurring in tropical and subtropical regions, identified currents and environmental selection as key to shaping its global biogeographic patterns[20]. The other study found that differences in microbial community composition at given stations in an ocean basin are correlated with transport time of the water masses[21]. However, we still lack a comprehensive understanding of how prokaryotic communities disperse by large-scale oceanic currents. The convergence of water masses by currents in frontal regions may lead to mixing of and concentrating prokaryotic communities[22], fractions of which may originate from different water masses. Depending on the environmental conditions and population size, and the adaptability of populations, distinct fractions of a community may preferentially grow in these mixing zones. Hence, we hypothesize that the composition of prokaryotic communities in these mixing zones consist of subcommunities or modules, i.e., groups of organisms whose source region can be traced based on a comparison with those communities in adjacent water masses.

Based on these assumptions, the aim of our study was to investigate how ecological mechanisms and large-scale oceanic current systems shape prokaryotic microbial community patterns in the global oceans with a particular focus on the Atlantic and Pacific Ocean. Further, we wanted to elucidate whether these communities exhibit a modular structure. Therefore, we developed a framework, based on co-occurrence network analyses, to identify environmentally distinct clusters of prokaryotic communities and examined how they disperse and mix in ocean basins by current systems. We define i) a cluster as a technical term for a densely interconnected group of network nodes as outcome of the co-occurrence analysis; ii) a module as a cluster of network nodes that represent groups of organisms with overlapping distribution patterns and iii) a subcommunity as a group of organisms that originate from the same source region and share similar ecological niches. We applied this framework and these terms to ASVs of the V4-V5 region of the 16 S rRNA gene in the upper 200 m along latitudinal transects in the Atlantic and Pacific Ocean from subantarctic to subarctic (Pacific) and temperate (Atlantic) regions (Fig. 1a). Further, we tracked these modules in other ocean regions using data sets of the Malaspina expedition and a transect across the entire Mediterranean Sea and analyzed their temporal dynamics using a multi-year study at the San Pedro Ocean Times series (SPOT) location in Californian coastal waters of the Pacific Ocean. To analyze the impact of hydrography on their distribution, we modelled ocean surface currents using a set of 25,000 globally distributed drifters. Our analysis shows that prokaryotic communities in the global oceans can be grouped into coherent modules of co-occurring ASVs whose distribution and mixing follows large-scale current systems.

## Results

### Prokaryotic communities in the Atlantic and Pacific Ocean: composition and shaping mechanisms

In total we sequenced and analyzed 814 samples along the two latitudinal transects in the Atlantic ($n = 409$) and Pacific Ocean basins ($n = 405$) with a mean sequencing depth of 21,791 counts per sample. For a more detailed insight into sequencing statistics, we refer to their original publications[8,9]. Along the Atlantic transect between 62°S and 47°N, covering all biogeographic provinces, we detected 1703 prokaryotic ASVS and along the Pacific transect between 52°S and 59°N, also covering all biogeographic provinces and various water masses[23], 1714 ASVs that were sufficiently abundant to surpass our abundance filter (Fig. 1a, Supplementary Fig. 1). Both ocean basins shared >90% of ASVs in the FL (0.2–3.0 μm) and PA prokaryotic communities (3–8 and >8 μm) (Supplementary Fig. 1). About 3% of the ASVs were unique to the Atlantic and about 6% to the Pacific Ocean. In the Atlantic Ocean, the unique ASVs occurred mainly beyond 50°S in the upper epipelagic (20 m to the deep chlorophyll maximum (DCM)) in the 0.2-3 and 3–8 μm size fractions and in the lower epipelagic (DCM to 200 m) in the >8 μm size fraction (Supplementary Fig. 2). There, the FL fraction expressed high abundance of *Gammaproteobacteria* unique to the region, whereas the >8 μm fraction showed mainly *Bacteroidetes* taxa that were unique to the Atlantic Ocean basin along the complete transect. In the Pacific Ocean unique ASVs occurred mainly in the equatorial upwelling where we detected *Planctomycetacia* and *Cyanobacteria*. In addition, in the PA size fractions we detected increased abundance of *Gammaproteobacteria*, in particular *Xanthomonadaceae*, that were unique to the south Pacific subtropical gyre (Supplementary Fig. 2). In most regions the PA size fractions showed higher abundance of unique ASVs than the 0.2–3 μm size fractions. Species richness in the Pacific Ocean was highest around 25–30° in both hemispheres (Fig. 1c). In the south Atlantic, species richness was highest around 30–35°S but in the northern hemisphere highest richness was skewed towards more tropical regions, indicating that this transect touched only the eastern edge of the subtropical gyre but also the upwelling region northwest of Africa. Lowest richness occurred in the Atlantic in the southernmost and in the Pacific in the northernmost subpolar regions. Considering a more balanced diversity measure taking into account richness and evenness, the effective number of species (ENS), the Atlantic and Pacific Ocean exhibited pronounced differences. In the Pacific, ENS was highest in the equatorial region whereas in the equatorial Atlantic it exhibited a minimum and pronounced maxima around 30° in both hemispheres (Supplementary Fig. 3). As the equatorial upwelling is much stronger in the Pacific than the Atlantic Ocean[24] the different ENS may reflect these basin-specific features.

To examine which ecological mechanisms contribute to prokaryotic community assembly and shape biogeographic patterns in both oceans we applied a phylogenetic framework based on null models to our data[12]. In the Atlantic Ocean, selection accounted for the largest fraction of the three mechanisms shaping biogeographic prokaryotic community patterns in the three size fractions and contributed 50–70% of identified mechanisms (Fig. 1b). For the FL and 3–8 μm-PA fraction, homogeneous selection was the most important mechanism followed by heterogeneous selection and dispersal limitation. For the >8 μm fraction, dispersal limitation was of equal importance as homogenous selection. These data are generally in line with the results from the Pacific Ocean (Fig. 1c; for more details see ref. 9). However, in the Pacific Ocean, heterogeneous selection was of consistently lower and dispersal limitation even of highest importance for the fraction >8 μm (Fig. 1c). These results are in contrast to the results of the Malaspina expedition which was restricted to subtropical and tropical regions and a depth of 3 m, and which identified drift as the most important mechanism[3].

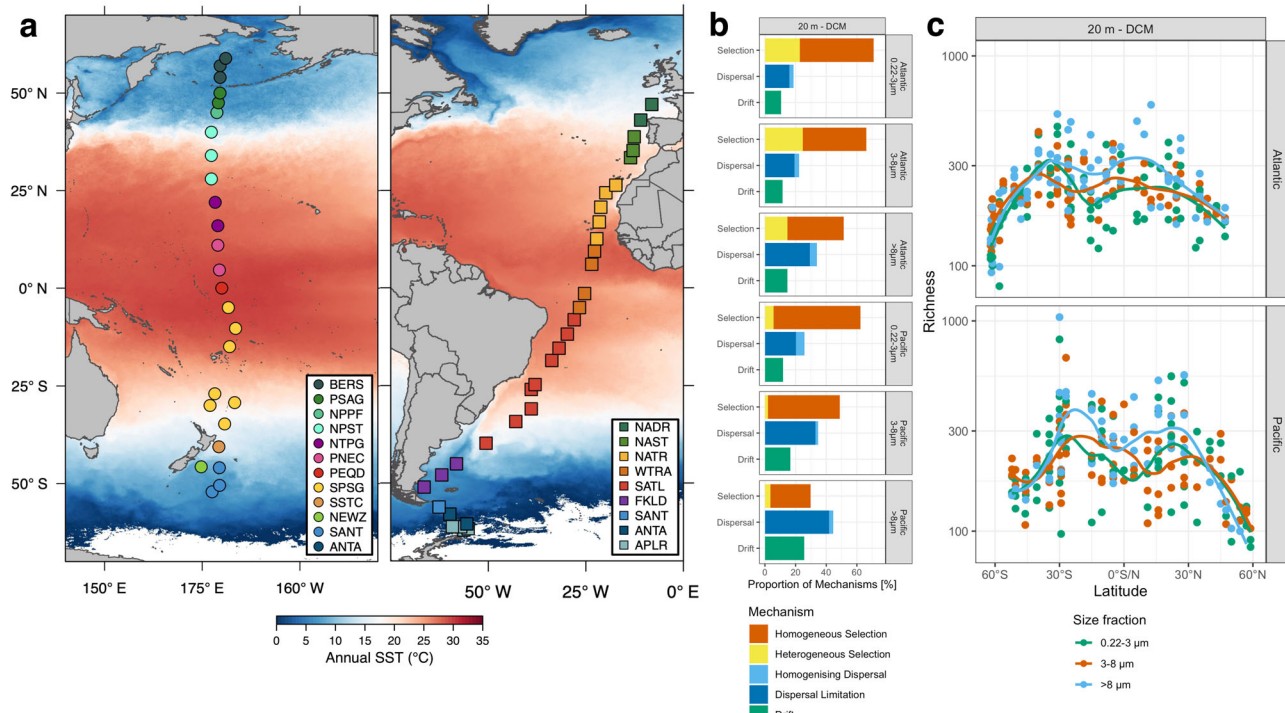

**Fig. 1 | Stations in the Atlantic and Pacific Ocean visited for assessing the prokaryotic microbiome, its richness and relative significance of the ecological mechanisms selection, dispersal and drift for community assembly. a** Stations visited in the Atlantic between 62°S and 47°N and in the Pacific Ocean between 52°S and 59°N. The color code of the stations refers to their affiliation to bio-geographic provinces according to Longurst[81] in the Atlantic (North Atlantic drift (NADR), North Atlantic subtropical gyre (NAST), North Atlantic tropical gyre (NATR), Western tropical Atlantic (WTRA), South Atlantic gyre (SATL), Southwest Atlantic Shelves (FKLD), Subantarctic water ring (SANT), Antarctic (ANTA), Antarctic polar (APLR)) and the Pacific Ocean (BERS: Bering Sea; PSAG: Pacific sub-arctic gyre; NPPF: north Pacific polar frontal region; NPST: north Pacific subtropical gyre; NPTG: north Pacific tropical gyre; PNEQ: Pacific north equatorial counter current; PEQD: Pacific equatorial divergence; SPSG: south Pacific subtropical gyre; SSTC: south subtropical convergence; NEWZ: New Zealand coastal province; SANT: subantarctic province). Stations are overlaid on maps of both oceans with annual mean sea surface temperatures (SST) (https://oceandata.sci.gsfc.nasa.gov). Maps are modifications of versions published previously according to CC BY[8,9]. **b** Relative proportions of homogeneous and hetero-geneous selection, homogenizing dispersal and dispersal limitation and drift on assembly of the 0.2–3 μm, 3–8 μm and >8 μm prokaryotic communities in the epipelagic (20 m to DCM depth) of the Atlantic and Pacific Ocean. **c** Richness of the ASVs of the V4-V5 region of the 16 S rRNA gene of the three size fractions of the prokaryotic communities in the epipelagic Atlantic and Pacific Ocean. Points indicate single samples and the line the smoothed mean value. Source data are provided as a Source Data file.

In the lower epipelagic Atlantic, homogeneous selection and dispersal limitation were of equal importance for shaping the FL prokaryotic community composition (Supplementary Fig. 4). For the 3–8 μm fraction, dispersal limitation was most important, followed by drift, and for the >8 μm fraction both ecological mechanisms were of equal importance. In the Pacific Ocean, drift was most important for both PA fractions and homogeneous selection for the FL prokaryotic community (Supplementary Fig. 4).

The relative importance of the ecological mechanisms with increasing temperature difference of the biogeographic regions differed between the Atlantic and Pacific Ocean. In the upper epipelagic Atlantic, the importance of homogeneous selection decreased with increasing temperature difference whereas that of heterogeneous selection increased (Supplementary Fig. 5b). Dispersal limitation was most important at intermediate temperature differences. Its importance continuously increased with temperature difference in the lower epipelagic for the PA fractions except when heterogeneous selection was most important (Supplementary Fig. 5b). The Pacific Ocean exhibited a decreasing importance of homogeneous selection and increasing impact of dispersal limitation with increasing temperature difference, more pronounced in the upper than in the lower epipelagic (Supplementary Fig. 5a, for further details see[9]).

We tested whether the distribution of samples within ocean basins affected the outcome of our analyses by downsampling both transects to cover the same latitudinal span (51 °S to 45 °N) and an equal

distribution of samples (Supplementary Figs. 6, 7). The downsampling strongly reduced the importance of heterogenous selection in both ocean basins, most probably due to the removal of subpolar samples. Further, the importance of homogenous selection was very low in the PA fractions of the Pacific Ocean especially compared to the Atlantic Ocean data. Despite these differences, overall patterns of ecological mechanisms were retained especially for 0.22–3 μm fraction where selection dominated prokaryotic community assembly.

## Modular structure of prokaryotic communities

Dispersal of prokaryotes via oceanic currents and mixing may counter separation of prokaryotic communities in different water masses thus contributing to the formation of communities, more broadly adapted to different environmental niches than source-communities. Such mixing may occur in particular in frontal regions or upwelling systems but may also affect the large central oceanic gyres, e.g. by mesoscale up- and downwelling. We thus hypothesize that in pelagic environments, a prokaryotic community at a given location consists of different modules of prokaryotes that are best adapted to the prevailing environmental conditions from their source regions. The composition of such communities is driven by a balance between physical mixing, dispersal and selective turnover rates of community members. We tested this hypothesis via inferring modules, termed clusters in a technical sense, on the basis of their co-occurrence topology: Prokaryotes that occur at similar environmental conditions form

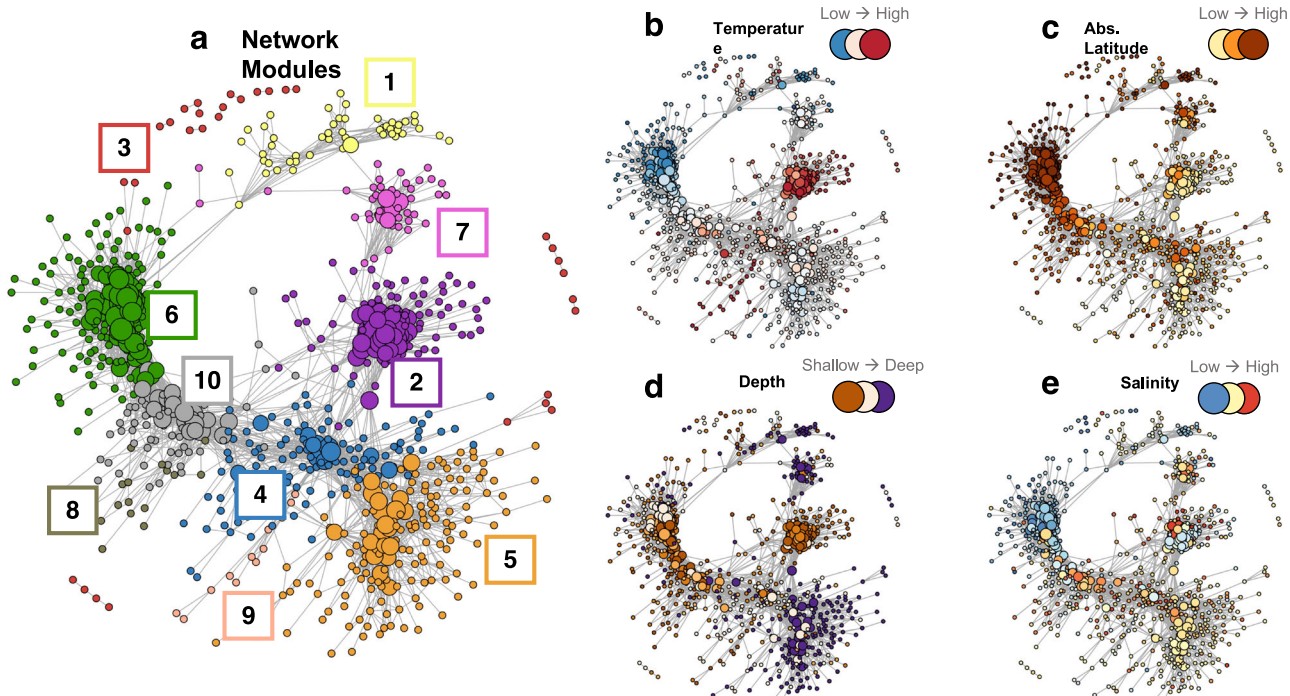

**Fig. 2 | Co-occurrence network clusters of the ASVs of all samples analysed in the Atlantic and Pacific Ocean. a** Co-occurrence network clusters with color codes of modules 1 to 10 used in Figs. 3–5. Nodes represent ASVs and node-size is relative to the number of edges of the respective node. Network associations were calculated with SparCC. **b** Network with continuous color scale for ambient temperature of stations and depths of the ASVs between 4 and 30 °C. **c** Network with continuous color scale for absolute latitude between 0 and 62°. **d** Network with continuous color scale with depth between 20 and 200 m. **e** Network with continuous color scale for salinity between 32 and 38.

interconnected clusters in co-occurrence networks that inherit their common niches with a distinct environmental signature (Supplementary Fig. 8). The network topology we inferred from the combined dataset of all Pacific and Atlantic samples showed distinct clusters of ASVs, i.e. modules, using the edge-betweenness algorithm (Fig. 2a). Colored network nodes according to the environmental conditions at the ASV's highest abundance revealed that these modules indeed show coherent environmental signatures regarding temperature, salinity, absolute latitude and depth (Fig. 2b–e) which matches the high explanatory potential of interaction adjusted beta-diversity indices such as TINA in the Pacific Ocean data[9]. We characterized the environmental niches by calculating the weighted mean of environmental variables using module-abundances as their weights. In total, we identified 10 modules with greatly varying diversities and relative abundances (Supplementary Fig. 9, Table 1). Modules 2, 4, 8 and 9 reflect warm water niches. They comprise but are not limited to *Oxyphotobacteria*, mainly *Prochlorococcus*, various orders of *Alphaproteobacteria* (SAR11, *Puneispirillales*, *Rhodospirillales*), various orders of *Gammaproteobacteria*, the euryarchaeotal Marine Group II and correlate with high salinity, high oxygen, low silicate and nitrate concentrations (Supplementary Fig. 8). Modules 3, 6 and 10 are very diverse, comprise Marine Group II, *Flavabacteriales*, SAR11, *Puneispirillales*, *Rhodobacterales*, SAR86 and SAR406 and reflect cold-water niches with high chlorophyll fluorescence and high silicate and nitrate concentrations (only modules 3 and 6). Modules 1, 5 and 7, comprising various lineages of *Gammaproteobacteria*, *Nitrososphaeria* and *Thermoplasmata* and prominent in the lower epipelagic, feature uniform environmental preferences including high silicate concentrations and prokaryotic bulk generation times and low chlorophyll fluorescence and prokaryotic cell numbers (for a complete list of ASVs allocated to each module see Supplementary data S1). Even though quite a few modules encompass ASVs affiliated to similar genera, the ASVs of each module are distinct.

The variations in the relative abundances of the different modules along the transect in both oceans indicate that the upper and lower epipelagic harbor communities of different modules adapted to different environmental conditions (Fig. 3a). A high overlap of modules and highly diverse communities existed between 30° and 40° in both hemispheres whereas in the tropics and subtropics more uniformly structured communities with a strong dominance of module 2 were present. In the Pacific Ocean, the module distribution from the tropics to subpolar regions was highly symmetrical in both hemispheres for all size classes. In the upper epipelagic, two major subcommunities dominated: Module 2 assigned to warm-water and module 6 assigned to cold-water conditions. Module 1 with higher relative proportions in the PA fractions, was consistently present along the transect without major latitudinal differences. In the mid-latitudinal transition zones, module 2, 6, 8 and 10 constituted enhanced proportions. Module 9 was restricted to the equatorial upwelling region in the upper epipelagic. The same modules were also present in the Atlantic Ocean except module 9 (Fig. 3a). However, instead of a symmetrical distribution poleward from the tropics in both hemispheres, the modules in the northern hemisphere overlapped over a larger latitudinal span and did not separate as distinctly as in the Pacific Ocean. A reason for this asymmetry may be that our Atlantic transect did not touch the central north Atlantic subtropical gyre but was located further east close to the Canary Islands and the adjacent regional upwelling systems.

The lower epipelagic of both oceans exhibited only a weak, if any latitudinal covariation of relative module abundances, presumably reflecting the rather uniform environmental conditions in this layer (Fig. 3a). Modules 1, 5 and 7 greatly dominated but with different proportions in both oceans and among the FL and PA fractions. The most striking differences were a much higher proportion of module 7 in the Atlantic and the dominance of module 1 in the PA fractions in the Pacific Ocean. These results indicate that basin-specific features of the prokaryotic communities are much more pronounced in the lower

**Table 1 | Description of modules**

| Module | 1 | 2 | 3ᵃ | 4 | 5 | 6 | 7 | 8 | 9 | 10 |
|---|---|---|---|---|---|---|---|---|---|---|
| No. of ASV | 49 | 129 | 32 | 113 | 165 | 175 | 46 | 16 | 12 | 76 |
| Marine Group II | | + | | + | + | + | | | + | + |
| Marine Group III | | | | + | + | | | | | |
| Nitrosopumilales | | | | + | + | + | | | | |
| Synechococcales | | + | | + | | + | | + | + | |
| Flavobacteriales | | + | | | | + | + | + | | + |
| SAR11 | | + | | + | | + | | | | + |
| Puniceispirillales | | + | | | | + | | | | + |
| Rhodospirillales | | + | | | | + | | | | |
| Rickettsiales | | + | | | | | | | | |
| Rhodobacterales | | + | | | | + | | | | + |
| Caulobacterales | | | | | | + | | | | |
| Sphingomonadales | + | | | | | | | | | |
| Alteromonadales | + | | | | | | + | | | |
| Ectothiorhodospirales | | | | | | | | | | + |
| Pseudomonadales | | | | | | | + | | | |
| SAR86 | | + | | | | + | | | + | + |
| SAR202 | | | | + | + | | | | | |
| SAR324 | | | | + | + | | | | | |
| SAR406 | | | | + | + | | | | | + |
| Abbreviation | PAL-C | TrSU-C | TEM-C | TrSLL-C | ULE-C | TSP-C | APL-C | SFU-C | PEU-C | TRP-C |
| Regions and depth range | Pacific Atlantic lower epipelagic | Tropic-Subtropic-upper epipelagic | Temperate- | Tropic-Subtropic Low-Light | Ubiquitous lower epipelagic | Temperate-SubPolar | Atlantic-Pacific-lower epipelagic | Subtropic-Frontal-upper epipelagic | Pacific Equatorial upper epipelagic | Temperate-Polar Frontal |

Modules, numbers of ASV per module, prominent prokaryotic orders of each cluster, abbreviation, region and depth range of each cluster. For a complete list of ASVs of each module see Supplementary Data S1.
ᵃvery diverse without any dominant orders.

than in the upper epipelagic and more so in the PA fractions, matching to reduced connectivity between communities in higher depths[15]. Interestingly, even though the lower epipelagic of the Atlantic and Pacific Ocean harbored distinctly different subcommunities of prokaryotes with completely different ASVs the dominant modules 1 and 7 were phylogenetically more closely related to each other than to any other module (Fig. 3d). Generally, the weighted UniFrac distance between modules correlated with the environmental preference of these modules along the transects, an indication of the phylogenetic signal found previously for prokaryotes in marine environments[3,9].

We characterized modules by analyzing its realized niche space built with a set of environmental parameters that were significantly improving constrained correspondence analysis (CCA). All 13 different environmental parameters used for CCA had a significant effect on model performance, as tested with forward selection. Module scores were distributed mainly along two axes (temperature and depth), spanning a triangle with three distinct corners: warm & low depth (module 2 and 9), cold & low depth (module 6), high depth (module 5) (Supplementary Fig. 10). To test how well the niche space of modules corresponds to respective niche space of ASVs, we ran the same CCA on ASV data. Their scores in the CCA biplot reproduced the triangular distribution of module scores and coloring ASVs according to module assignment perfectly and fit to the distribution of modules uncovered with the CCA built on module distribution (Supplementary Fig. 11). The abstraction of groups of ASVs into modules hence retained their realized niche preferences and further explained a much higher amount of inertia of the respective datasets (modules explaining 46.9% and ASVs 13.4% of inertia).

To test the general validity of this network cluster framework we tracked ASVs with module assignments in the data of the Malaspina expedition, which used the same primer set as our study. We were able to identify these modules also in this data set, comprising up to 80% of total prokaryotic abundance, and found that over large regions of the Pacific, Atlantic and Indian Ocean module 2 greatly dominated (Fig. 4a). At the northernmost stations of the Atlantic, the southernmost stations of the Pacific and in various upwelling regions in the Indian Ocean, e.g. west of Australia, module 8 constituted relatively highest proportions. Module 6 occurred only in the upwelling regions west of Australia and west of South Africa. Interestingly, module 9, restricted in our data set to the equatorial Pacific, was also detected as exclusive to this region. This ocean harbored other subcommunities in its equatorial region, which were not covered by our data set.

The application of our framework to a data set of a transect from the eastern to the western Mediterranean Sea, applying also the same primer set[25], revealed a clear transition from the dominance of module 2 in the eastern to that of module 6 in the western region in the upper 20 m (Fig. 4b). In the adjacent Atlantic Ocean, also covered by this transect, module 8 dominated, in agreement with the other data sets. At the deep chlorophyll maximum modules 4 and 6 greatly dominated.

To validate the temporal robustness of our network cluster analysis we applied it to a multi-year period of SPOT in southern California coastal waters of the Pacific Ocean, also using the same primer set[26]. The results show a clear dominance of module 6 from 2005 to 2018 in the FL (0.2–1.0 μm) as well as the PA fraction (1–80 μm) of prokaryotic communities with seasonal variations (Fig. 5a, b). In agreement with the data from the Atlantic and Pacific transect module 5 dominated at 150 m depth (Fig. 5b). A phase from mid-2014 to early 2016 with a dominance of warm-water module 2 coincided with a strong maximum of the Oceanic Niño Index (ONI)[27], reflecting a strong El Niño event (Fig. 5a) that was previously described to alter microbial community composition[28]. We found that the ratio between warm-water module 2 and cold-water module 6 was significantly different during the El Niño

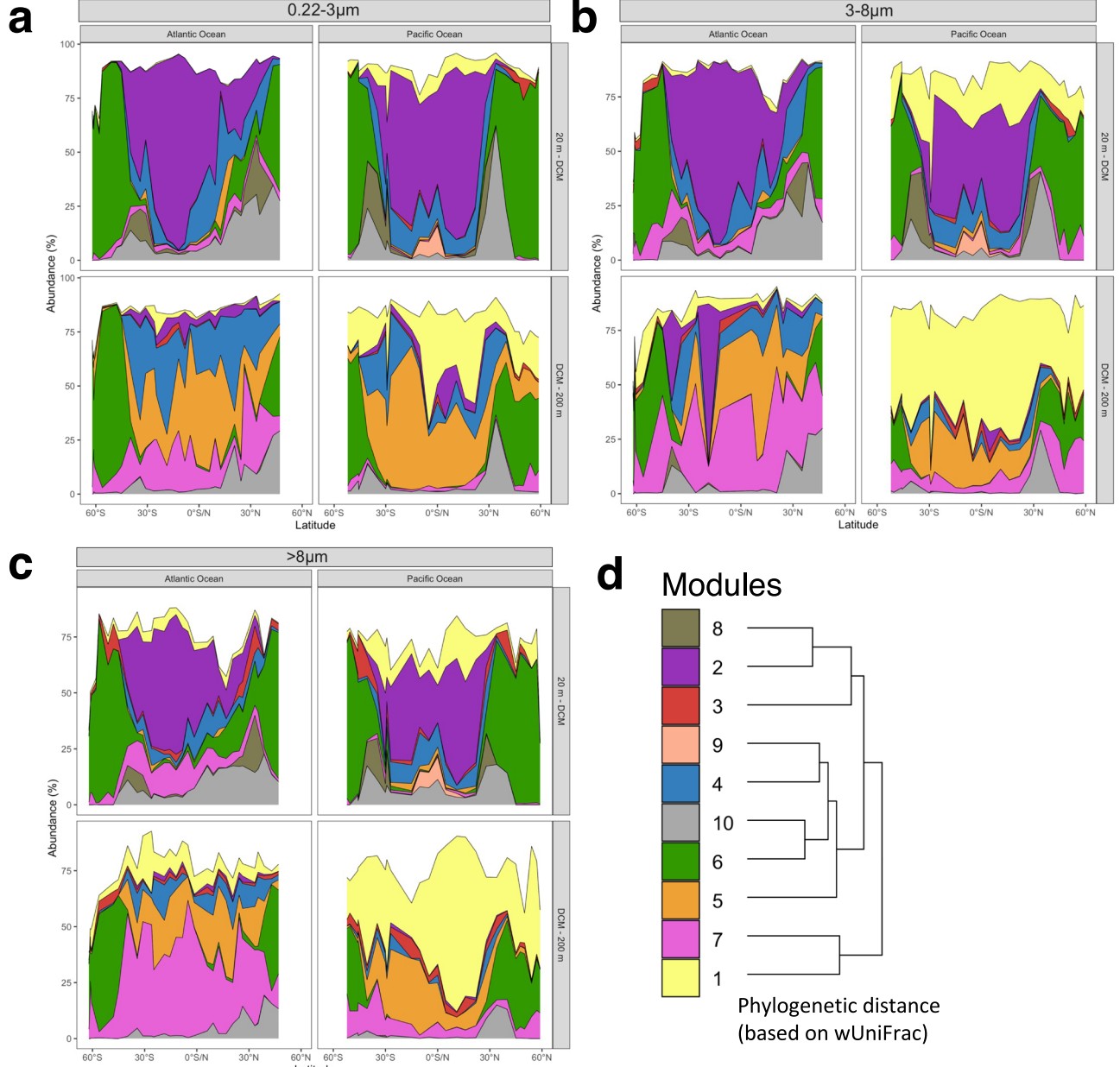

**Fig. 3 | Prokaryotic communities structured into co-occurrence modules in the epipelagic Atlantic and Pacific Ocean. a** Distribution of modules of the 0.2–3 μm size fraction, encompassing free-living prokaryotes, between 62°S and 47°N in the Atlantic and between 52°S and 59°N in the Pacific Ocean in the upper (20 m-DCM) and lower epipelagic (DCM-200 m). For the color code see **e. b** Distribution of modules of the 3–8 μm size fraction encompassing particle-associated prokaryotes. **c** Distribution of modules of the >8 μm size fraction encompassing particle-associated prokaryotes. **d** Phylogenetic distance based on weighted UniFrac analysis of modules 1 to 10. Source data are provided as a Source Data file.

event as compared to outside that event ($p$-value $= 3.5 \times 10^{-9}$) hence validating the biological significance of the inferred modules.

These results indicate that our network cluster analysis is suitable for the application to other data sets and to trace temporal as well as spatial variations in the composition of prokaryotic communities and to identify advection of water masses carrying residing prokaryotic subcommunities/modules as indicators such as during an El Niño event. We named modules based on the ocean basins, latitudinal regions and depth ranges in which the clusters predominantly occurred, as indicated in Table 1.

**Current systems affect community dispersion**

We hypothesize that the distribution of modules along the latitudinal transects in the Atlantic and Pacific Ocean establishes via an interplay between selective and dispersal processes, mainly controlled by the balance of microbial and current system-related timescales on which these mechanisms act. To analyze the modular structure of local communities, we subset the global co-occurrence network into local networks constituted of single samples by filtering out only those nodes whose ASVs were present in the individual samples. We analyzed how well local communities could be dissected into individual modules by calculating the modularity of the communities and relating changes in network modularity to spatial positions along the latitudinal transects in the Atlantic and Pacific Oceans. In our data, modularity also indicates the presence of multiple network clusters within a single sample, hence emphasizing mixing of subcommunities at positions along the transects where we detected high modularity (Figs. 6a and S12). Modularity peaked at subtropical latitudes between 25 and

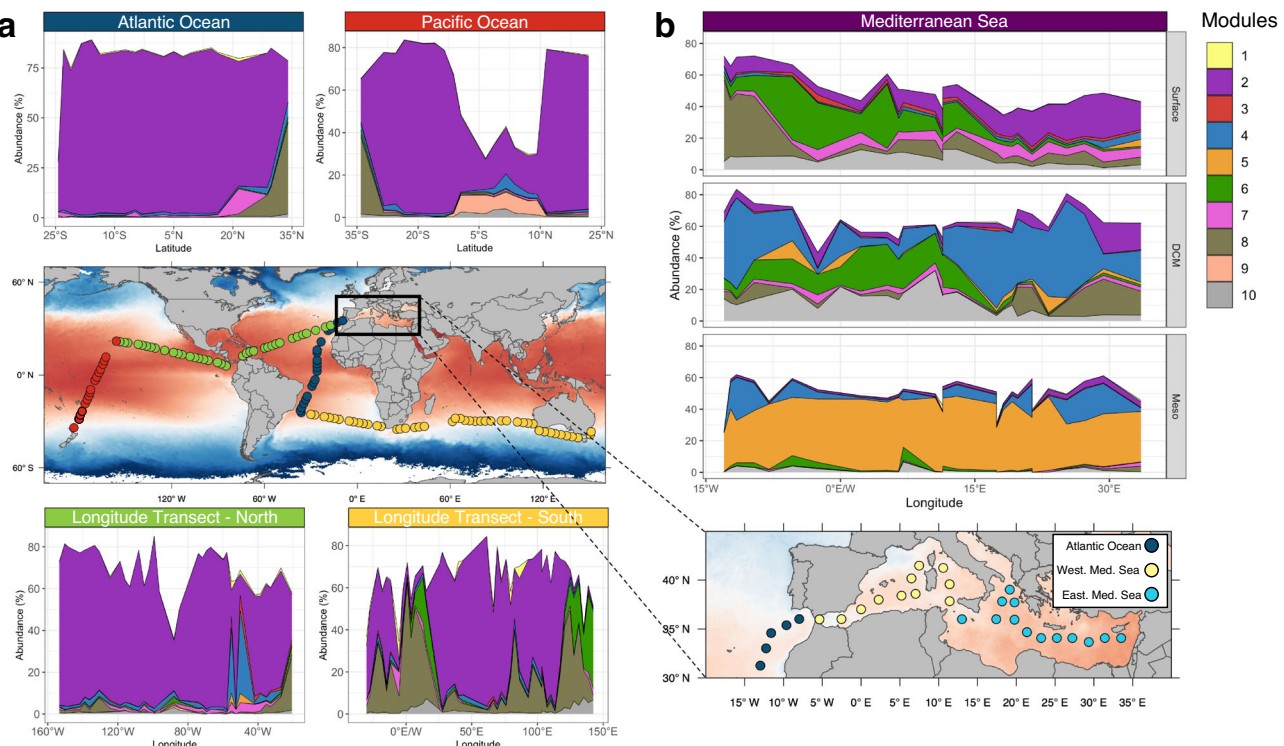

**Fig. 4 | Prokaryotic communities structured into co-occurrence modules in the Pacific, Atlantic and Indian Ocean and the Mediterranean Sea. a** Modular structure in latitudinal and longitudinal transects in the near surface Pacific, Atlantic and Indian Oceans based on data collected at the indicated stations of the Malaspina expedition at 3 m depth[3]. **b** Modular structure in an east-west transect across the Mediterranean Sea and the adjacent Atlantic Ocean at the surface, the DCM and in the mesopelagic (pooled depths 200-1000 m). For further details on the stations see reference Sebastian et al. [25]. Source data are provided as a Source Data file.

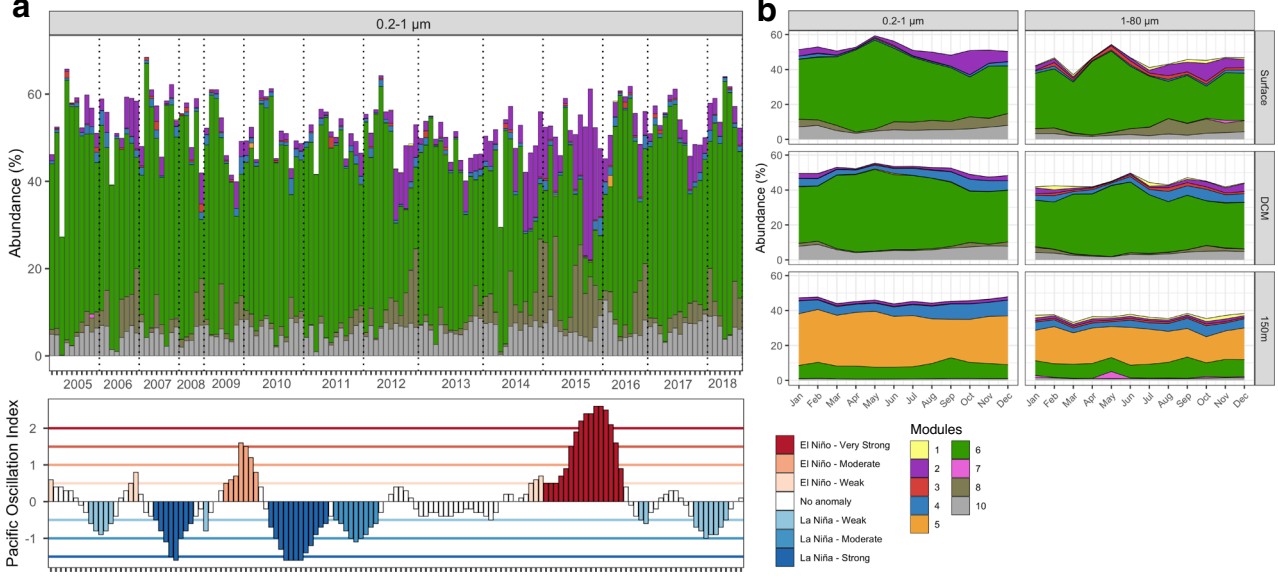

**Fig. 5 | Prokaryotic communities structured into co-occurrence modules at the San Pedro Time series (SPOT) location in Californian coastal waters of the Pacific Ocean from 2005 to 2018. a** Modular structure of the 0.2-1 μm size fraction at the surface and the Oceanic Niño Index (ONI) from 2005 to 2018. The color code of ONI indicates La Niña and El Niño events. **b** Annual patterns of the module distribution of the 0.2–1 and 1–80 μm size fractions at the surface, the DCM and at 1000 m depth. Monthly data were calculated as means of each month from the entire time series. Data are taken from ref. 26. Source data are provided as a Source Data file.

35° in both hemispheres in the epipelagic of both oceans with an additional peak in the equatorial region of the Pacific Ocean. The latitudinal modularity patterns matched the latitudinal diversity gradient that we observed before. We tested whether the number of simultaneously occurring modules linearly correlates with richness of prokaryotic communities and found significant linear relationships for all size-fraction, depth-layer and ocean basin combinations (Supplementary Fig. 13).

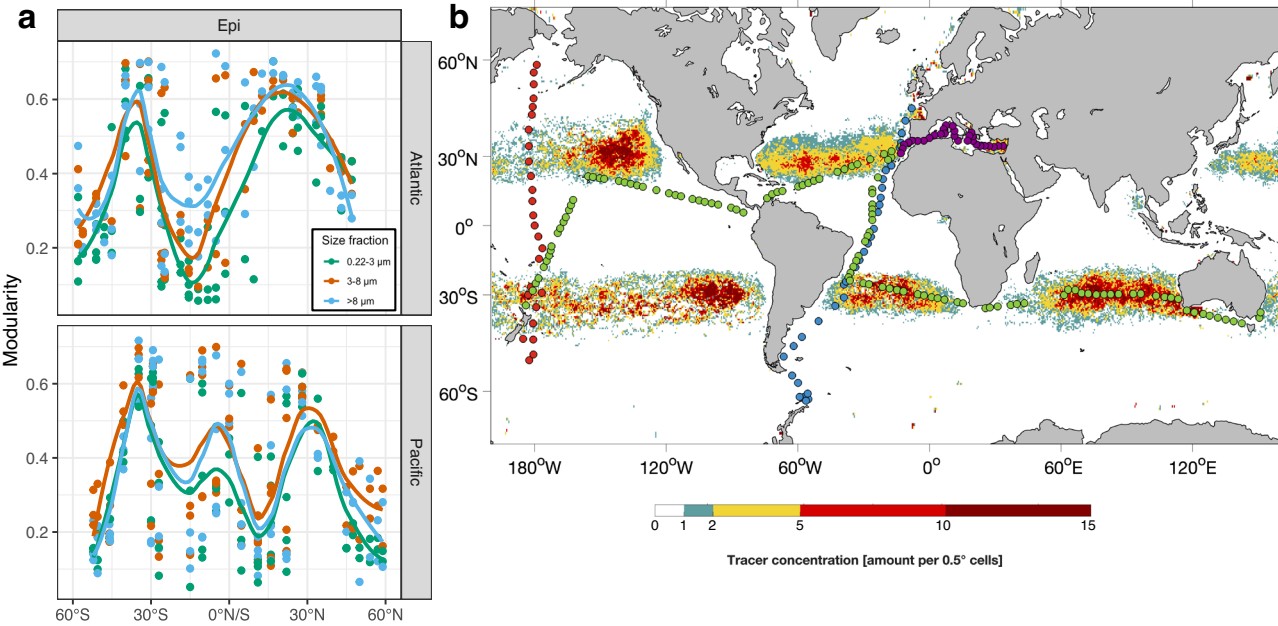

**Fig. 6 | Modularity of prokaryotic communities in the Atlantic and Pacific Ocean and tracer concentration in the global oceans after a five years modeling simulation. a** Modularity of the prokaryotic communities of the 0.2–3, 3–8 and >8 μm size fraction in the Atlantic between 62°S and 47°N and in the Pacific Ocean between 52°S and 59°N in the upper epipelagic (20 m-DCM). Points indicate single samples and the line the smoothed loess-fit. **b** Tracer concentrations in the global oceans as a result of a five year simulation of drifter movement. Points indicate

stations of all cruises considered in this study; red: Atlantic Ocean (RV Polarstern cruises ANTXXVIII/4 and /5, ref. Milke et al. [8]), blue: Pacific Ocean (RV Sonne cruises SO248 and SO254, ref. [9]); green: Malaspina expedition (ref. [3]); violet: Mediterranean Sea (RV Sarmiento de Gamboa, ref. [25]). For tracer distribution patterns at 0, 1, 2, 3, 4 and 5 years see Fig.S6). Source data for subplot a are provided as a Source Data file.

Assuming that these patterns emerge as a result of the major currents and transport of water masses in the ocean basins, we simulated dispersal of flotsam in the oceans, using a dataset of 25,000 drifters. We calculated aggregation patterns within the global oceans based on van der Mheen et al. [29]. The simulation of drogued drifters over five years, tracing transport of surface water, revealed approximate accumulation areas which correspond with regions of high modularity of epipelagic prokaryotic communities around 25-35°N/S adjacent to the subtropical fronts (Fig. 6b, Supplementary Fig. 14). In the equatorial regions, tracer concentrations decreased rapidly, indicating a permanent and strong advection of water masses towards subtropical regions. We further analyzed the absolute mean velocity of drifters and found globally structured patterns such as strong currents over 75 cm/s in equatorial regions and at eastern continental coastlines (Supplementary Fig. 15). Absolute mean velocity was minimal in accumulation areas where it dropped below 5 cm/s, underlining that accumulation areas are formed by the combination of convergent currents and reduced velocity. From this analysis it is also apparent that analyzing the impact of large-scale currents on prokaryotic communities is most suitable along the latitudinal axis, crossing prominent velocity gradients, compared to the longitudinal axis, where rather homogenous current velocities were found.

Modularity in the lower epipelagic was high between 30°S and 30°N in both oceans and only decreased beyond (Supplementary Fig. 12). The high modularity in the upper epipelagic of the equatorial Pacific Ocean appears to reflect that diverse subcommunities from the lower epipelagic and upper mesopelagic reached these near surface layers mixing with residing communities. Within northern and southern subtropical regions where trade-winds persist, wind driven Ekman currents dominate the dispersion of tracers towards the subtropical gyres, roughly located at 30°S/N, adjacent to the subtropical fronts[22,30,31]. In these regions, prokaryotic subcommunities from low chlorophyll warm-water regions mix with cold-water associated

subcommunities from high chlorophyll polar frontal regions. Hence, enhanced richness and modularity appear to be a result of the mixed and regionally entrapped water masses, similar to the accumulation of microplastic[32,33]. In the tropical regions of the gyres of the Pacific and south Atlantic Ocean and in the polar frontal regions and beyond, modularity was low (Fig. 6a). This indicates reduced mixing of different subcommunities, presumably due to high homogeneous selection and reduced dispersal limitation (see above). We inferred tracer concentrations in a regular grid around each sampling location for various simulation times. Tracer concentrations were highest in accumulation zones and perfectly matching to regions of highest modularity and diversity of modules (Supplementary Fig. 16).

## Discussion

Our study provides a systematic comparison of prokaryotic communities in the Atlantic and Pacific Ocean between subantarctic and subpolar/boreal regions. Both oceans share more than 90% of the ASVs and some general latitudinal biogeographic patterns considering richness, ENS and module distribution. However, despite this high similarity our results reveal basin-specific differences, more so in the PA fractions and in the lower epipelagic than in the FL fraction and the upper epipelagic, respectively. Differences were most pronounced in two regions: the southernmost Atlantic, i.e. the Atlantic sector of the Southern Ocean, not covered by the southernmost stations in the Pacific Ocean; and the equatorial upwelling including the south subtropical gyre in the Pacific Ocean, regions with basin specific hydrographic conditions. These results corroborate previous findings of the Malaspina expedition, which showed that in the tropical and subtropical oceans prokaryotic communities within basins were more similar than between basins, a feature more pronounced in the epipelagic than in the bathypelagic[4,34]. Our analyses expand these findings to a systematic coverage of the entire basins of the Atlantic and Pacific Ocean and also include different size fractions.

Studying the effect of current-induced dispersal on prokaryotic communities, well known in freshwater microbial ecology (e.g[35–37]), has attracted less attention in microbial oceanography[38] and its potential has not been fully explored. Hydrographic models were used to model the effect of dispersal on eukaryotic plankton based on Tara Ocean data[20], to study prokaryotic community assembly in the East China Sea[39], to data sets of the Malaspina expedition[3] and to a latitudinal transect in the Pacific Ocean[9]. These studies examined ecological mechanisms for community assembly but did not consider that microbial communities are composed of coherent groups other than phylogenetic components and that these groups represent sub-communities with different ranges of occurrence. The modular structure of prokaryotic communities and ecological mechanisms for their assembly have not been explored in an ocean basin nor in the global oceans.

Our results show that the co-occurrence cluster analysis is most suitable for such an investigation. The applied SparCC network inference builds associations between ASVs based on both, putative biological interactions and common environmental preferences. It yielded sound results indicating that the prokaryotic communities in the global oceans are composed of groups of ASVs with overlapping spatial distribution, here called modules, each of which has different ranges of occurrence and is distinct in different latitudinal and depth ranges. Co-occurrence inference requires a sufficient prevalence of ASVs, which limits our analysis to the rather abundant part of prokaryotic communities (representing 90% of total reads). We were able to identify ten distinct modules with different ranges and preferred environmental conditions of occurrence, such as in tropical, subtropical, temperate or subpolar regions or in the upper and lower epipelagic (Figs. 3–5, Table 1). We further validated these modules by running randomization analyses to make sure they are not formed by chance. Another study that inferred a global co-occurrence network based on the Tara Ocean dataset found five distinct modules that were differential abundant in three latitudinal regions[40]. They applied an inference tool that minimized correlation between ASVs based on mutual environmental preferences, unlike the approach that we used to infer sub-communities of ASVs with similar ecological niches. In our case, most modules exhibit a global coverage and were detected in environmentally similar regions of the Atlantic, Indian and Pacific Ocean as well as in the Mediterranean Sea. The relative partitioning differed between FL and PA community fractions, in ocean basins and along latitudinal (Atlantic, Pacific Ocean) and longitudinal transects (Atlantic, Pacific, Indian Ocean, Mediterranean Sea). Differences were more pronounced in the lower epipelagic and the PA fractions, indicating more basin-specific features in these size classes and the lower than in the upper epipelagic and the FL prokaryotic fraction. These findings are in line with the results from Sommeria-Klein et al.[41] that analyzed biogeographic patterns of eukaryotes of various sizes and found latitude specific biogeographic patterns in smaller sized organisms and basin-specific patterns in larger sized organisms. The long-term data at SPOT in Californian coastal waters further indicate that the modules are a consistent and prevailing feature of oceanic prokaryotic communities and that their temporal dynamics reflect seasonal oscillations and even an El Niño event[28]. Therefore, based on the temporal stability of the inferred module composition, we assume that the distribution of modules is associated to large-scale current systems that express only marginal seasonal variation[18]. Interestingly, module 9 (PEU-C) with the lowest abundance of all modules was detected only in the upper epipelagic equatorial Pacific (Figs. 3c, 4a), suggesting that this region harbored a unique subcommunity, specifically reflecting the biotic and environmental conditions of this major upwelling system. In the Atlantic, Indian and Pacific Ocean the modules we identified made up more than 75% of the total prokaryotic community abundance in most regions, indicating that our approach captured the abundant majority of the community members and

modular structure. However, the data of the Mediterranean Sea, SPOT, the equatorial Pacific close to the surface at 3 m depth from the Malaspina expedition and the subantarctic Atlantic covered lower proportions of the communities (down to 30%), suggesting that other, so far unidentified modules contribute substantially to community assembly. Since the module associations of ASVs was built only upon the data from the two latitudinal transects in the Atlantic and Pacific Ocean, covering open ocean communities below 20 m depth, we were surprised to find high abundance of modules even in other ocean regions, such as the Mediterranean Sea and in the coastal environment of SPOT.

We hypothesize that the simultaneous occurrence of distinct modules in given regions of the global oceans with different ranges of occurrence reflect different niche realizations in the Hutchinsonian sense[42]. Our reconstruction of the realized niche space using ASVs revealed distinct niches that clustered according to module assignments and analogue analyses using module distribution showed that module scores reflect the position of these clusters in niche space (Supplementary Figs. 10 and 11). The inferred network modules preserved ASV scores in niche space and further improved model performance using a set of environmental parameters by threefold. We assume that the coexistence of the two modules 1 & 7 with highly similar environmental preferences reflects dispersal limitation between the two oceans together with historical contingency and other seemingly stochastic effects that control community assembly at higher depths[9]. The realized niches of (micro-) organisms are dynamic, they evolve due to evolutionary adaptation[43] and converge with niches of biologically interacting partners[44]. Simultaneously, dispersal due to oceanic currents leads to an expansion of range limits of organisms that dwell in regions with strongly divergent currents[45]. Hence, we assume that the observed clustering of realized niches in the CCA biplots reflects the interplay of the beforementioned processes, leading to distinct groups of co-occurring organisms that we defined as modules.

The modules we identified by the co-occurrence analysis encompass rather different numbers of taxa, between 12 to 175 distinct ASVs (Table 1, Supplementary data S1), and the modules are differentially linked (Fig. 2). Most phylogenetic orders are present in only a few or even one module. As we targeted ASVs with relatively long 16 S rRNA gene sequences (411 bp), our taxonomic resolution can be on the species or subspecies level for certain taxonomic groups. Several phylogenetic orders such as *Flavobacteriales* or SAR11, and even genera such as *Prochlococcus*, occur in different modules, however, represented by different ASVs. As most modules occur in several oceans this implies that ASVs of a given module occur in different oceans at similar environmental and biotic conditions. On the other hand, different ASVs of a given order or genus in a given region may belong to different modules, implying that they co-occur with different taxa. These features of the different modules presumably reflect the fine-tuned adaptation of a given taxon, represented by an ASV, to the environmental and biotic conditions and selection on the (sub-) species level. It has been shown that these fine-tuned adaptations exist for *Prochlorococcus* and the SAR11 clade and involve functional genes, enabling the taxa to occupy different niches and expand their niche space[6,9,46–48]. Further, temperature ranges and means of taxon-specific functional genes of prokaryotic species differ along a latitudinal transect in the Atlantic Ocean[6] and affinities of nutrient uptake systems of marine pelagic prokaryotic species vary[49–51]. Hence, the different co-occurring modules may reflect that different species or subspecies of a given genus or species, respectively, originate from different regions with different ecological niches and are dispersed by large-scale current systems. This implies that communities are composed of modules from various source regions, a feature remaining unnoticed in a classical prokaryotic diversity analysis. These features may also explain why large-scale taxonomic distance-decay relationships differ from

those of functional genes and general functional categories, exhibiting great functional redundancies on the latter level but little on the functional gene level[1,6,52]. Different ASVs of a given taxon that occur simultaneously can have different module assignments, each inheriting biotic interactions with other organisms associated to the same modules rather than across modules. This would explain why interaction adjusted indices such as TINA perform so well in explaining community variation compared with classical compositional indices[9,53]. Differences may also reflect the different life styles, i.e. as FL or PA bacteria as shown for the different composition and partitioning of modules 1, 5 and 7 in the lower epipelagic (Fig. 3a-c).

We identified homogeneous selection as the major ecological mechanism for community assembly in the upper epipelagic of the Atlantic and Pacific Ocean and dispersal limitation as second most important. In the lower epipelagic, dispersal limitation was of equal importance as homogeneous selection except in the Pacific Ocean. With increasing temperature difference, dispersal limitation became increasingly important (Supplementary Fig. 5), reflecting that water masses of different temperature and harboring different communities as occurring in frontal regions prevent or limit mixing and thus dispersal. Even though homogenizing selection was identified as the dominant mechanism, presumably most pronounced in large and rather homogeneous water masses, dispersal limitation does play a role in such systems as shown in the Southern Ocean[54]. Heterogeneous selection was shown to be relevant in the Atlantic Ocean, indicating that community composition in the subantarctic regions diverged strongly from that further north. Drift was least important except in the PA fractions in the lower epipelagic Atlantic and Pacific, indicating that stochastic processes contribute only little to community assembly in the upper epipelagic and act mainly on the PA communities. It must be kept in mind that PA prokaryotes do not constitute more than 20%, and usually less than 10% of the total prokaryotic communities in oceanic systems[23,55,56]. This emphasizes that deterministic processes shaping predominantly the FL prokaryotic communities are generally of much greater importance for the community assembly of the major oceanic biogeochemical players. Assembly of PA prokaryotic communities appears to be controlled differently and more by stochastic mechanisms including mesoscale up- and downwelling, nutrient pulses, phytoplankton bloom development and subsequent sinking of the senescent and aggregated phytoplankton[57–59]. Our findings are in contrast to those of Logares et al. [3], based on the Malaspina dataset and restricted to the tropical and subtropical global oceans. These authors found that drift was the major mechanism and homogeneous selection and dispersal limitation were of equal importance. The Malaspina expedition sampled along both, latitudinal and longitudinal transects that included many samples from tropical regions and hence a high proportion of sample comparisons between samples from the same latitudinal ranges. As our subsampling approach showed (Supplementary Figs. 6, 7), the sampling strategy can have a strong effect on the analysis outcome. Therefore, we want to highlight that our analysis of ecological mechanisms reflects the importance of mechanisms acting along the latitudinal gradient. Our findings extend those of Milke et al. [9] from the Pacific to the Atlantic Ocean and underscore that deterministic processes are the major agents shaping the structure of prokaryotic communities in the major oceans, when considering entire ocean basins.

Our results imply that the deterministic ecological mechanisms, selection, both homogeneous and heterogeneous, dispersal limitation and homogenizing dispersal, do not always act on the entire prokaryotic community but may act preferentially on specific modules. Clusters of co-occurring organisms imply that distinct taxa, here ASVs, are affected by common biotic and/or environmental variables. Hence, depending on the environmental or biotic conditions homogeneous or heterogeneous selection or homogenizing dispersal may favor one or another module as shown for the shifts in the module partitioning in

the latitudinal or longitudinal oceanic transects or the time series at SPOT.

The simulation of drifter movement by surface currents showed a rapid advection of water masses away from the tropics and to accumulation of tracers near 25-30°N/S in the Atlantic and Pacific Ocean (Fig. 6b, Supplementary Figs. 14–16), a result of near-surface Ekman drift and geostrophic currents[31]. In these regions, modularity of local prokaryotic communities and number of simultaneously occurring modules were particularly high, corresponding with highest ASV richness along the transects. This finding suggests that in these subtropical gyral regions, adjacent to the subtropical fronts, accumulation and mixing of prokaryotes belonging to different modules, ranging also further north or south, takes place. Here, dispersal limitation and mixing of different water masses in the frontal regions presumably led to this accumulation, the relatively high number of co-occurring modules, enhanced modularity and high richness. In the equatorial upwelling region of the Pacific ENS and modularity of the prokaryotic community were also enhanced, presumably a result of the continuous supply of nutrients and prokaryotes from the mesopelagic. On the other hand, the central gyral regions with no import of foreign water masses, except randomly and rarely by regional upwelling, exhibited a reduced richness and a lower number of modules with the dominance of the tropical-subtropical epipelagic module 2 (TrSU-C), suggesting that homogeneous selection was the dominant acting ecological mechanism. An enhanced richness near the subtropical front in the South Pacific and at the equatorial upwelling has been reported previously[5]. We do not only confirm these findings but show that they are a general feature occurring near all major subtropical frontal regions in the Pacific and Atlantic Ocean and that they are a result of enhanced mixing of adjacent water masses harboring differently structured prokaryotic communities. Interestingly, the regions adjacent to the subtropical fronts also accumulate microplastic particles, indicating that they behave similarly as prokaryotes, and passively disperse[22,30,33]. The most sophisticated analysis of latitudinal diversity gradients in marine microorganisms to date did not resolve diversity peaks in subtropical regions and found temperature to be the most important driver of diversity[2]. Different to our and previous studies that detected increased diversity at subtropical fronts[5,7], Ibarbalz et al. used the Shannon index to quantify alpha-diversity instead of richness which is less biased by differential detection of rare ASVs. However, it cannot resolve the total number of observable features which rather fits to our hypothesis of increased diversity due to mixing of different subcommunities. Our analyses further show that the interplay of deterministic ecological mechanisms, predominantly homogeneous selection and dispersal limitation, global current systems and water mass advection shape the structure of oceanic prokaryotic communities.

Geographic distance has been used in several studies to assess biogeographic patterns of prokaryotic communities in ocean basins and globally[1,4,6,9]. Surface current systems have only rarely been considered in such studies, which would enhance our understanding of the connectivity of water masses and thus potential dispersal pathways. To the best our knowledge, there is only one study, which specifically tested for the effect of large-scale currents on basin-scale biogeographic patterns and the composition of pro- and eukaryotic plankton communities[21]. These authors based their analyses on the Tara Ocean dataset and found an inverse relationship between the prokaryotic community composition and the minimum travel time of up to 1.5 years. We applied a heuristic approach to identify regions with convergent waterflow based on a global dataset of drifters. It yielded approximate regions where drifter accumulate with increasing simulation time. The resulting tracer (virtual drifter) concentrations increase in accumulation regions that we matched against our biological data. Our approach indicates that dispersal would advect prokaryotes and flotsam within two to three years from the equatorial and

the subpolar regions to the polar frontal and subtropical frontal regions. This time span is far higher than the mean generation time of the entire prokaryotic community, which ranges from <1 to 15 days between the tropics and subpolar regions of the Pacific Ocean[23]. It underscores that community turnover through homogeneous selection and dispersal limitation are the main drivers to maintain the composition and modular structure of the prokaryotic communities in any biogeographic region, fueled by input of energy and substrates from primary production.

Our results show that the composition of the microbiome of the Atlantic and Pacific Ocean exhibits great similarities but biogeographic patterns feature also basin-specific characteristics between subantarctic and subarctic/boreal regions of both oceans, mainly in the PA fractions and more so in the lower than in the upper epipelagic. Our approach of the co-occurrence network analysis yielded valuable and unprecedented insights into the modular structure of oceanic prokaryotic communities, not only in the Atlantic and Pacific Ocean but also in the Indian Ocean and the Mediterranean Sea and thus globally. Homogeneous selection and dispersal limitation were identified as the major ecological mechanisms for assembly of the prokaryotic communities and the major current systems were shown to play an important role in assessing the biogeographic patterns, community assembly and modular structure. The approaches to analyse the co-occurrence networks and the local modularity presumably are applicable to other ecosystems such as lakes, rivers and even the human gut. Dissecting the structure of the residing prokaryotic communities in these ecosystems into subcommunities will certainly shed new light on their composition and assembly.

## Methods

### Sample collection
Data to assess the prokaryotic communities in the Atlantic and Pacific Ocean were taken from two data sets of samples which were collected during research cruises with RV Polarstern in the Atlantic and RV Sonne in the Pacific Ocean. These cruises spanned latitudinal transects from 62°S to 47°N in the Atlantic and from 52°S to 59°N in the Pacific Ocean. Samples were collected with 10- or 20 l-Niskin bottles mounted on a CTD-rosette at 20, 40, 60, 100 and 200 m depth and additionally from the depth of the DCM if not covered by one of the standard depths. In total, 61 stations (33 Atlantic, 28 Pacific) were sampled, yielding in total 891 samples. Water samples were filtered sequentially through 8 μm (mixed cellulose ester SCWP14250, Millipore, Darmstadt, Germany), 3 μm (mixed cellulose ester SSWP14250, Millipore) and 0.22 μm (polyethersulfone GPWP14250, Millipore). Filters were stored at −20 °C until further processing. For further details of sampling, cruise track and location of stations in the Atlantic and Pacific Ocean see Milke et al.[8] and[9]. In addition, a suite of environmental variables was assessed such as temperature, salinity, oxygen, chlorophyll fluorescence, prokaryotic abundance and growth. For further details on the methods to analyse these variables see[6] for the Atlantic and[23] for the Pacific Ocean.

### DNA extraction and library preparation
Thawed filters were cut into pieces and transferred into a 2 ml tube containing garnet beads (0.7 mm) and stored at −20 °C until DNA extraction. Total genomic DNA was isolated from each filter using a chemical and mechanical lysis method by applying first bead beating to the filter pieces. For the Atlantic Ocean, we used the Mo-Bio Ultra-Clean Soil DNA Isolation Kit (MO-BIO Laboratories Inc., Carlsbad, CA, USA) and for the Pacific samples the DNeasy PowerSoil Kit (Qiagen GmbH, Hilden, Germany). This kit uses the same chemistry as the former Mo-Bio kit but was renamed after MO-BIO Laboratories Inc. was taken over by Qiagen.

The V4-V5 hypervariable region of the 16 S rRNA gene (515 F: GTGYCAGCMGCCGCGGTAA −926R: CCGYCAATTYMTTTRAGTTT)

was PCR-amplified according to[60] as outlined in Milke et al.[8,9]. Briefly, we applied a combinatorial primer pair which included Illumina-adapter sequences, Illumina-primer sequences and barcode/index-sequences for multiplexing[61] using the same amplification procedure. Amplified DNA was cleaned using 0.8X Ampure XP magnetic beads (Beckman Coulter, Brea, CA, USA) and quantified using a Qubit device with the dsDNA HS kit (Thermo Fisher). Samples were diluted to equal DNA concentration, pooled and cleaned again using the SPRIselect magnetic beads (Beckman Coulter). Cleaned pooled samples were sequenced on an Illumina MiSeq using the PE300 chemistry and demultiplexed in the sequencing center of the Helmholtz Centre for Infection Research (Braunschweig, Germany).

### Bioinformatic pipeline
Demultiplexed sequences were processed after Yeh et al.[61] following the scripts at github.com/jcmcnch/eASV-pipeline-for-515Y-926R. First, primer-sequences were removed using cutadapt (version 3.3) allowing an error rate of 20% to retain sequences with mismatches to the primer. Since the V4-V5 primer pair is able to amplify 16 S and 18 S sequences, the sequences were split accordingly using bbsplit from the bbtools package (version 38.7) by comparing the sequences against a curated 16 S/18 S database built on SILVA132[62] and PR2[63]. The 16 S sequences were further processed in qiime2 Version 2019.7[64] by first cutting bad quality ends of the sequences (250 forward and 220 reverse reads) and subsequently denoising the sequences using the DADA2 plugin in qiime2[65]. The DADA2 plugin included a quality-filtering for which we used the default settings, a sequence-merging step and a chimera removal step. Finally, reads were classified using the classify-sklearn plugin in qiime2 against a SILVA132 database which was subset to the primer region. All sequences classified as plastids (chloroplasts and mitochondria) were removed, retaining only prokaryotic 16 S sequences. All sequences with a classification confidence of smaller than 0.8 were removed from the dataset. ASV tables were exported from qiime2 and further processed using R (version 3.6.3)[66].

### Abundance filter
We applied an abundance filter to remove rare ASVs which are likely to produce spurious correlations during network inference. We filtered prokaryotic ASVs individually for each ocean basin according to Milici et al.[7] who used a conservative abundance-filter that retained ASVs that were present in an abundance >0.001% of the whole dataset and (1) were present in at least one sample at a relative abundance >1%, or (2) were present in at least 2% of samples at a relative abundance >0.1% of a sample, or (3) were present in at least 5% of samples in any abundance. Implications of this filter setting on diversity metrics and biogeography are analyzed in Milke et al.[8].

### Phylogenetic tree calculation
For analyses based on phylogenetic measures, we constructed a phylogenetic tree by first aligning all prokaryotic sequences against a SILVA138 alignment using SINA (version 1.2.9). The phylogenetic tree was calculated with FastTree based on the global alignment (version 2.1.11). Cophenetic distances between ASVs were calculated using the stats package in R.

### Ecological mechanisms
To quantify the relative importance of the ecological mechanisms selection, dispersion and drift, we applied a phylogenetic framework based on null-models[9,12]. Briefly, we calculated phylogenetic distances between samples based on their ßMNTD[67] and evaluated their significance by comparing the distances against their respective values of 999 shuffled datasets. A significant phylogenetic effect was considered if the |ßMNTD| was more than 2 standard deviations larger than the mean value of all shuffled instances. In that case, >+2 standard deviations indicated heterogeneous selection and <−2 standard deviations

indicated homogenous selection. If no significant phylogenetic effect was detected, we tested for a significant compositional turnover between samples by calculating the Raup-Crick index adjusted for relative abundances[68] by creating 999 accordingly shuffled datasets. The difference between observed Bray-Curtis distance and its respective distances in the shuffled datasets, $RC_{bray}$, was scaled from −1 to +1. An $RC_{bray} > 0.95$ is defined as dispersal limitation, $RC_{bray} < -0.95$ is defined as homogenizing dispersal and $|RC_{bray}| < 0.95$ did not indicate a significant dispersal effect. In that case, hence $|\beta MNTD| < 2$ and $|RC_{bray}| < 0.95$, the differences between samples cannot be distinguished from neutral processes and therefore defined as stochastic drift.

## Network inference and analysis

We used the SparCC algorithm[69] to infer co-occurrences of ASVs based on correlation values from compositional data. Hence, the filtered ASV table was split into coherent subsets according to the sampled ocean, size-fraction and depth-layer resulting in 12 distinct data subsets. We applied the SparCC inference according to[69] on the unrarefied count data of each sample subset and bootstrapped inference by shuffling the dataset and calculating the correlation matrix 999 times. The p-value was defined as the proportion of bootstrapped correlation values for which a correlation value at least as extreme as the one computed for the unshuffled data was found[69]. For further analysis, all positive correlations with a correlation value of ≥0.5 and a p-value adjusted for multiple comparisons of ≤0.05 were considered. Network analysis was conducted using the R package igraph (version 1.5.1). Therefore, all network-subsets were combined to produce one global undirected network of positive associations. We excluded negative associations from our network as we focused on clusters of simultaneously occurring organisms that are formed by either beneficial interactions among organisms, or by mutual environmental preferences. Network topography was plotted using a force-directed layout and node-size was depending on the number of associations per node. For an approximate characterization of network layout, we colored all nodes according to the environmental properties of the sample with the maximum abundance of the corresponding ASV. Cluster association was determined by applying edge-betweenness[70]. We extracted subnetworks of individual samples by filtering the nodes of the global network to the respective ASVs present in each single sample. Subnetworks were further analyzed using igraph by calculating modularity (using *modularity* function). As clusters, modules and subcommunities are interchangeable terms, we defined them as follows: A cluster is a technical term for a densely interconnected group of network nodes. A module is a network cluster in its biological context. A subcommunity is a group of organisms that originate from the same source region and share similar environmental preferences.

To assure that modules represent significant features that are not formed by chance, we created 999 shuffled instances of the network by defining randomly terminal nodes for edges at a probability of 0.5 using the rewire function of the igraph package. For each shuffled network, we applied edge-betweenness clustering and compared the resulting modularity of the cluster object against the observed modularity. We defined the p-value for our null hypothesis that the observed network modularity results from random network topology as the proportion of modularity values from shuffled networks that are at least as extreme as the observed value (p-value < 0.001). Since significance of network-edges are based on bootstrapping, we assume that network properties such as number of nodes and edges represent true features that are not formed by chance. We further verified the defined modules by testing how much variability of the count data is explained by modules by applying permutational MANOVA using Bray-Curtis transformed count data and 999 permutations. The module assignment for each ASV explained significant amount of variation ($p < 0.001$, $F = 20.556$, $DF = 10$ of 820 in total).

## Testing module validity

We tested the validity of our inferred modules by assigning the corresponding module number to each sequence identifier from the qiime exported feature table. As these identifiers offer unique codes for each sequence, we could use them to search for module members in other datasets that were run through the same bioinformatic pipeline. Therefore, we downloaded nucleotide data from large scale ocean sampling projects that applied the same primer we used, the Malaspina expedition[3], a longitudinal transect in the Mediterranean Sea[25] and a 12 year time series recorded at SPOT in southern Californian coastal waters[26]. To test for an effect of El Niño Pacific Oscillation on the occurrence of modules in the SPOT time series data, we exported the Oceanic Niño Index (ONI) from the NOAA webservice (https://www.cpc.ncep.noaa.gov/data/indices/oni.ascii.txt) for the months present in the time series and rated its relative intensity from <−2.5 to >2.5 after[71].

## Statistics

All analyses were performed with R (R Core Team 2021, version 4.1.2). Maps were plotted with package oceanmap (version 0.1.3) and the underlying annual mean sea surface temperature (SST) data were retrieved from MODIS-Aqua[72]. Richness and effective number of species (inverse Simpson) were computed with the package vegan (version 2.6-4) after rarefying samples to equal sequencing depth of 8000 counts per sample. We chose this threshold as sufficiently deep to display the prokaryotic diversity while not losing too many samples due to insufficient sequencing depth. While richness of microbial communities is not a robust measure of alpha-diversity[73], it does show the total number of inherent phylogenetic features and hence fits to our working hypothesis of increased diversity at regions of community mixing, for which abundance-weighted diversity measures, such as Shannon Entropy, are less applicable. All other analyses were done using unrarefied count data (SparCC analysis) or relative abundance data (inference of ecological mechanisms, PERMANOVA). Weighted UniFrac and Bray-Curtis dissimilarity were computed using inhouse scripts. The tree based on weighted UniFrac dissimilarities was calculated with the *hclust* function (method = "mcquitty") and displayed with the ggtree package (version 3.2.1). The environmental preferences of each module were computed as the mean of environmental parameters, weighted by module abundance within each sample. The corresponding values were normalized by total sum for each parameter and standardized with Z-scores for each module. Standardized and normalized values were plotted in a heatmap, with rows and columns ordered according to their hierarchical clustering.

We created a model to explain sample variation based on a set of environmental parameters using automated stepwise model building in forward direction with the *ordistep* function of the vegan package. The model was built using constrained correspondence analysis (CCA). We displayed significant environmental parameter in the biplot of the final model together with species scores (here ASV scores). The same analysis was done with module abundance data instead of using ASVs, displaying module scores together with sample scores in the final biplot.

## Drifter calculations

Tracer concentration calculations were derived from the Global Drifter Program dataset of 24,971 Surface Velocity Program (SVP) drifters[74] deployed from 1979 until March 2020. The drifter data were quality-checked, the raw position data were interpolated at 6-hour intervals and the data product was provided by the National Oceanic and Atmospheric Administration (NOAA)[75]. The SVP drifters consist of a spherical surface float containing the telemetry unit and a "holey-sock" drogue with a length of ~6 m centered at 15 m water depth to reduce the direct wind slip on the surface float. Loss of the drogue may be caused by the mechanical forces constantly acting on the surface float and drogue connection. In the case of a drogue loss, a sensor detects

the absence of the drogue and the data of drogued and undrogued drifters can be distinguished. For the tracer concentration calculations, only drogued drifters were considered to avoid any potential bias caused by wind and wave-induced currents. Since the average lifetime of an SVP drifter is one year[74], it is not possible to gain insight into the long-term advection and accumulation. To overcome this timescale limitation, we derived transport matrices from the drifter location data used in a number of studies to identify accumulation zones of marine debris[29,33,76].

To construct the transport matrix, the ocean is divided into rectangular cells with a cell size d$x$ x d$y$. The drifter trajectories are subdivided into subtrajectories with a length of d$t$. The initial and final cell of each subtrajectory and the number of subtrajectories starting in a given cell $i$ and ending in a specific cell $j$ are determined and normalized by the row sum of 1, which return a probability for each cell. The Probability Density Function (PDF) describes the probability that a tracer moves from one cell to another cell in the time interval d$t$. These PDFs are used as a proxy for the fraction of tracers that accumulate in specific cells. A detailed presentation of the advection using the transport matrix approach are provided by[29] and[77].

To derive the transport matrices, the entire dataset was gridded onto a d$x$ = 0.5° and d$y$ = 0.5° horizontal grid and the separation time interval was set to 30 days. The supporting information by van der Mheen et al.[29] contains a sensitive analysis of the choice of the grid resolution and the separation time interval, as well as technical details for the construction of transport matrices from the GDP dataset.

Simulations were performed for 1, 2, 3, 4 and 5 years to analyze the evolution of accumulation areas over time. The results for each year were plotted on a map which also indicates the position of the sampling stations using Matlab 2021a.

Tracer concentration data was gridded on the location data of the sampled stations for the latitudinal transects in the Atlantic and Pacific Ocean basin. Ten longitudinal cells (=5°) were averaged around each station to avoid bias in tracer concentrations because of undersampling due to limited drifter coverage.

**Reporting summary**
Further information on research design is available in the Nature Portfolio Reporting Summary linked to this article.

## Data availability
The nucleotide data used in this study is deposited at the European Nucleotide Archive (ENA) under the following accession numbers: PRJEB51015 (Pacific Ocean transect), PRJEB50983 (Atlantic Ocean transect), PRJEB48162 & PRJEB35673 (SPOT time series), PRJEB25224 (Malaspina data), PRJEB44474 (Mediterranean Sea transect). Environmental data of the Pacific and Atlantic Ocean transect are deposited at PANGEA[78,79]. Drifter data is available at the National Oceanic and Atmospheric Administration[80]. Source data are provided with this paper.

## Code availability
All R scripts to recreate figures and analyses were uploaded to Github and archived on Zenodo (https://doi.org/10.5281/zenodo.8273515).

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

## Acknowledgements

We are grateful to H. Hillebrand for comments on an earlier version of this publication. This work was supported by grants from the German Ministry of Education and Research (BMBF), BacGeoPac 03G0248A and PoriBacNewZ 03G0254A, and by Deutsche Forschungsgemeinschaft within the Collaborative Research Center TRR51 *Roseobacter*.

## Author contributions

F.M. and M.S. designed the study. F.M. carried out the analyses. F.M. and M.S. wrote the manuscript. J.M. carried out the analysis of the drifter data. All authors reviewed the final version of the manuscript.

## Funding

## Competing interests

The authors declare no competing interests.
