## [Peer Review File · Nature Communications]

Ecological mechanisms and current systems shape the modular structure of the global oceans' prokaryotic seascapeREVIEWER COMMENTS

Reviewer #1 (Remarks to the Author):

This work investigates the mechanisms shaping the structure of prokaryotic communities in the ocean, including marine current systems. In addition, the structuring of communities into modules is analyzed. Overall, the work is interesting and includes novelty. Yet, I have some comments and concerns mentioned below:

General comments:

-One aspect that should be analyzed/discussed more is how the sampling strategy could influence the measured ecological processes. The used "null model" approach (i.e., the "Stegen" method), is highly sensitive to spatial dimensions and the nature of the dataset. So, to what extent the measured processes can be extrapolated, or are representative? This needs to be tested/discussed, in my opinion.

Furthermore, to obtain robust measurements of ecological processes, different regions should be sampled more or less equally (i.e., if one environment experiencing similar environmental conditions is sampled more than others, this may bias the calculation of ecological processes towards the dominance of homogeneous selection, for example). By looking at figure 1B, it appears that tropical areas of the Atlantic in the transect were more intensively sampled than other areas. I suggest that the dataset is subsampled to contain a number of samples per environmental region (tropical, subtropical, temperate, and subpolar) that is proportional to the region size and that ecological processes are re-calculated. This would allow determining whether there are biases in calculating ecological processes linked to unequal sampling effort.

-Module validation: I think it would be good to test that modules are not obtained by chance. For that, modules could be tested in networks obtained from randomized data (i.e., a randomized ASV table) to determine their validity.

-Modules are obtained from association networks. They could reflect similar environmental preferences or ecological interactions. The manuscript could mention what these modules

are expected to be.

-It is unclear what is the contribution of the temporal data from SPOT to the main storyline and conclusions, including the comments related to "El Niño". This could be better integrated or just mentioned as a validation of the approach.

-Generally, the manuscript could be clearer (multiple sections are hard to follow).

-Substantial sections of the Discussion seem to be recapitulating results. For example, L328-370

Specific comments

L89-91: Clarify

L99-101: Clarify

L161-3: Hard to follow

L172-4: Clarify

L181-2: "Colored network nodes" Statistical values could be included

L233-6: Were these modules identical or similar? to what percentage?

L249: It is unclear what information is added by this analysis (SPOT). Is it just a validation?

L273-5: Clarify

L329-31: This should be validated, with, for example, randomization analyses that show that the obtained modules differ from chance.

L347-9: Does this refer to richness or abundance? If referring to richness, it would be surprising that most diversity is participating in modules

L351-2: Provide values for the lower proportions. Why should there be other modules? (additional taxa could be present and not be part of modules)

L355: "indicate that the oceanic prokaryotic communities exhibit a modular structure." Randomization analyses are needed to support this, in my opinion. That is, you probably need to show that random distributions of taxa do not lead to a modular structure.

L375-7: Unclear, clarify

L382-6: Unclear, clarify

L571: Indicate what edges mean. If using SparCC, edges may represent similar environmental preferences or ecological interactions. One way to assess whether edges could represent similar environmental preferences is using Flashweave, for example.

Reviewer #2 (Remarks to the Author):

In the submitted article, authors Milke et al examine V4-V5 16S rDNA amplicon sequences from 61 stations along two novel latitudinal ocean transects, from samples collected on the Malaspina expedition, and from the SPOT time series. They apply a correlation network analysis using the SparCC algorithm to identify co-occurring modules of taxa; use a combination of betaMNTD, the Raup-Crick index, and Bray-Curtis distances to estimate the relative importance of deterministic versus stochastic process on community assembly; and incorporate a large drifter dataset to estimate surface current transport matrices. The authors use genomic, bioinformatic, and statistical techniques commensurate with the standards of the field. Additionally, their results demonstrating global prokaryotic community modules that are largely structured by homogenizing selection, as well specific regions with communities that are strongly influenced by current-driven dispersal, are particularly exciting.

However, I struggled to understand the environmental and taxonomic significance of the modules identified by the authors. The description of the modules in the text was muddled and the naming scheme (Table 1) did not add clarity. As a result, the biogeography of the modules presented in Figure 3 did convey any impactful result that I could easily interpret. I would implore the authors to more carefully craft their description of the modules and pair that description with a consistent color and naming scheme throughout the figures and text.

Additionally, many of the arguments throughout the text and particularly in the Discussion were either worded poorly, had confusing structure, or were not well supported. I have endeavored to point out some of those instances in the minor comments below. I would encourage the authors to give this document a thorough examination to ensure that both the text and figures convey clear conclusions to the reader.

In sum, this analysis shows promise to advance the fields of marine microbial ecology and biogeography. However, the conclusions conveyed by the text and figures lack clarity and, thus, it is difficult to interpret their overall significance.

Minor Comments:

Lines 33-34: The structure of this sentence and the regions described is confusing. I am unsure whether additional commas are needed after “Indian Ocean” and/or “Mediterranean Sea.”

Lines 39-40: What does “concert the dispersal” mean?

Line 78: “these studies were missing” or “these studies missed”

Line 140: 50-70% of what?

Lines 146-147: “...Malaspina expedition, which was restricted to subtropical and tropical regions and a depth of 3 m, and which identified...”

Line 262: This seems like an odd place to finally name the modules. I would suggest including this information earlier when the modules are first being described.

Line 271: “relating”

Lines 281-284: Did the authors attempt to correlate tracer accumulation with modularity? According to the figure there are large areas with no overlap in estimates. However, it appears that the southern longitudinal transect has significant potential overlap between both metrics. Comparing the two metrics along this transect could be a powerful way to demonstrate a quantitative relationship between the authors’ two estimates of dispersal.

Lines 308-310: The language here is difficult to understand. Also, I believe there is a “i)” without a corresponding “ii)”

Line 315: “also include”

Lines 317-320: This claim is not well supported, there has been significant effort to characterize prokaryotic biogeography in the surface ocean.

Line 320: “It has been applied...” What “it” are the authors referring to here?

Lines 322-325: This is a run-on sentence. Please rephrase.

Line 329: Is “they” referring to co-occurrence cluster analysis?

Lines 353-355: This is circular reasoning. The authors used a statistical technique designed to identify co-correlated clusters of taxa, i.e. modules. Thus, is it unsurprising that “the oceanic prokaryotic communities exhibit a modular structure.”

Lines 375-379: How would the authors identify the “source region” where a theoretical bacterial species “originates” from? Could the co-occurrence of modules simply imply multidimensional niche partitioning of the biotope in the classical Hutchinson-ian sense (see for instance Colwell and Rangel 2009)?

Line 404: “do not constitute more than 20%, and usually less than 10%, of the total”

Line 406: Why are “ecological mechanisms” more important for free living prokaryotes? Is drift not an ecological mechanism?

Lines 454-456: This sentence structure is confusing, please rephrase

Line 483: delete “well”

Lines 481-485: Run-on sentence. Please split into two sentences.

Line 510: It is encouraging that 90% of the ASVs were observed in both ocean basins (Lines

116-124), but the differences in extraction technique between the Atlantic and Pacific samples could contribute to the 10% differences observed between the two basins. The authors should outline what differences exist between the two extraction kits. Alternatively, were any replicate samples extracted with each technique and compared?

Line 514: "Illumina"

Table 1: I find the names chosen confusing, as they do not appear to use a systematic structure. For instance, "lower epipelagic" is referred to with both an "L" (as in PAL-C) and with an "LE" (as in ULE-C). Similarly, the word "temperate" is referred to with "TEM" (TEM-C), "T" (TSP-C), and "TR" (TRP-C). Or, as another example, I am not sure what the difference is between "lower epipelagic" and "Low-Light."

Figure 1: I would encourage the authors to reconsider the layout of this figure. As currently configured it is hard to follow. Perhaps having the richness plots aligned latitudinally with their associated ocean basin map (with the proportion plots underneath) would assist?

Figure 2: In (b) on the figure move the "e" in "Temperature" up to a single line

Figure 3: I find it hard to interpret what the changes in the relative abundance of the modules mean from a taxonomic, environmental, or functional perspective. The authors might consider a color scheme denoting the module's environmental association (i.e., shades of red color for "Modules 2, 4, 8 and 9," shades of another color for "Modules 3, 6 and 10," and a final color for "Modules 1, 5 and 7").

Reviewer #3 (Remarks to the Author):

This manuscript uses two longitudinal transects in the Pacific and Atlantic oceans to construct prokaryotic species modules based on co-occurrence of ribosomal DNA variable region genes. The authors found sometimes similar modules in both oceans and relate their distributions with abiotic and biotic factors. Finally, they study the roles of selection and biased dispersal in their distributions, suggesting a strong role for current systems in module

distributions.

The use of new data sets along latitudinal transects is of interest, as well as the characterization of conserved modules. Their validation by external data, and some interpretations based on transport by currents are also of interest. However, the paper does not completely integrate this study in the current literature. For instance, it should be discussed with the results of Chaffron et al. (Science Adv 2021) regarding the modular structure of plankton communities, or with Sommeria-Klein et al. (Science 2021) for the effects of dispersion according to taxonomic groups and biogeography (for eukaryotes). It looks rather superficially to the influence of currents. The watermasses are not described according to their temperature/salinity characteristics, which are important for advection studies. It is to be discussed if latitudinal sampling is the best for studying current effects. The simulation with drifters may help for this, but it is unclear how the authors use it and reach their advection time of 2-3 years. More details and figures appear necessary for this analysis. More generally, some results do not seem to be based on clear statistical methods, that should be better explained and their limitations indicated.

The sequencing effort per sample is apparently not indicated. As the number of recovered ASVs seems relatively low, this should be clarified. What part of the diversity is accessed with this sequencing depth, and how does it impact modules? The analyses on ASVs seem to be done with relative abundances, do the authors also test the robustness of their conclusions using rarefactions?

Lines 114-135, it may be interesting to have more information about the taxa that are unique or common between basins. It is unclear if the results of the richness analysis along the latitudinal gradient is new compared to Ibarbalz et al.?

Line 233, the comparison with Malaspina data is of interest, can the authors discuss the potential effects of different sampling/sequencing methods and depths?

Line 255 and Fig5A, the correlation between module abundances and El Niño is not obvious, and may be due to random effects. A statistical analysis is necessary.

Line 225, it is well known that biogeography is more pronounced in surface than in deep samples, the literature should be better cited here.

Fig5B, it is interesting to see that the time series does not show large seasonal variations in module abundances. This seems to reinforce the conclusions, and may be further described.

Reviewer #1 (Remarks to the Author):

This work investigates the mechanisms shaping the structure of prokaryotic communities in the ocean, including marine current systems. In addition, the structuring of communities into modules is analyzed. Overall, the work is interesting and includes novelty. Yet, I have some comments and concerns mentioned below:

General comments:

-One aspect that should be analyzed/discussed more is how the sampling strategy could influence the measured ecological processes. The used "null model" approach (i.e., the "Stegen" method), is highly sensitive to spatial dimensions and the nature of the dataset. So, to what extent the measured processes can be extrapolated, or are representative? This needs to be tested/discussed, in my opinion.

Thank you for your generally favorable and supporting suggestions and for your constructive remarks helping us to further improve our analyses and manuscript.

Furthermore, to obtain robust measurements of ecological processes, different regions should be sampled more or less equally (i.e., if one environment experiencing similar environmental conditions is sampled more than others, this may bias the calculation of ecological processes towards the dominance of homogeneous selection, for example). By looking at figure 1B, it appears that tropical areas of the Atlantic in the transect were more intensively sampled than other areas. I suggest that the dataset is subsampled to contain a number of samples per environmental region (tropical, subtropical, temperate, and subpolar) that is proportional to the region size and that ecological processes are re-calculated. This would allow determining whether there are biases in calculating ecological processes linked to unequal sampling effort.

Regarding your valid point on our sampling strategy: Indeed, the null-model-approach is sensitive to the sampling region, but also to the phylogenetic diversity that is covered within the sample set. As suggested, we downsampled the dataset to cover equal number of samples in both ocean basins and further restricted the samples to the overlapping latitudinal range. This led to the removal of subpolar regions, which drastically reduced the phylogenetic diversity within the dataset. Consequently, the proportion of "heterogeneous selection" was much lower in the Atlantic Ocean dataset. In the Pacific Ocean basin downsampling resulted in a much lower importance of homogeneous selection in PA communities, an additional consequence of the reduced phylogenetic diversity due to the removal of subpolar samples. We added these results (l. 174-181) including an additional Figure to the supplement (Fig. S6 & S7).

-Module validation: I think it would be good to test that modules are not obtained by chance. For that, modules could be tested in networks obtained from randomized data (i.e., a randomized ASV table) to determine their validity.

Thank you for this important suggestion. We now validated the modules using three different approaches. First: SparCC association significances are derived from randomization of the original data, so that we can be sure that associations that we detected do not result by chance. Second: We randomly rewired network associations 999 times and calculated modularity scores that we compared with the modularity derived from the observed network. We found that the observed modularity was in all cases higher than the modularity from randomized networks. Third: We checked whether module-assignment explains a significant amount of ASV variation. For that, we ran a permutational MANOVA with module assignments as an explanatory variable. We found a highly significant impact of module assignment on ASV variation. A corresponding section was added to the Methods (l. 695-707).

-Modules are obtained from association networks. They could reflect similar environmental preferences or ecological interactions. The manuscript could mention what these modules are expected to be.

Thank you for this interesting suggestion. We tried to detect prokaryotic communities that consist of ASVs which co-occur for mainly two different reasons: Either they co-occur because they are biologically interacting, e.g., because of symbiotic relationship. Or they co-occur because they share a

highly similar environmental niche. The resulting network showed that groups of highly interconnected ASVs exist that form discrete (sub-)communities. To make clear that these (sub-)communities can reflect both, interacting organisms or organisms with similar env. preferences, we made an additional statement in the discussion (l. 381-382).

-It is unclear what is the contribution of the temporal data from SPOT to the main storyline and conclusions, including the comments related to "El Niño". This could be better integrated or just mentioned as a validation of the approach.

Our aim to include the SPOT data was to test whether the modules we identified are not only a spatial biogeographic feature but also apply to temporal features. The SPOT data set was the only one which is suitable for this analysis as it applied the same primer set as we did. We thus could not only confirm that the modules, in fact, exist but also that only minor seasonal and interannual variations occur at this coastal station over more than ten years. Further, this analysis showed that an El-Niño event leads to deviations from the long term seasonality. To clarify this we made an additional statement in the manuscript (l. 290-292).

-Generally, the manuscript could be clearer (multiple sections are hard to follow).

Thank you for this critique. We rephrased quite a few parts of the manuscript based on the more detailed reviewer comments below. You can identify them by the resubmitted version with track changes.

-Substantial sections of the Discussion seem to be recapitulating results. For example, L328-370

Thank you for this critique. We substantially modified this paragraph (and made changes at other sections) such that it contains now many new aspects elaborating on the results. Further, we added a comparison with appropriate references to this paragraph based on the comments from reviewer 3 (l. 380-422).

Specific comments

L89-91: Clarify
Done. (l. 89-91)

L99-101: Clarify
Done. (l. 98-102)

L161-3: Hard to follow
We rephrased this sentence and the entire paragraph (l. 174-181).

L172-4: Clarify
Done. (l. 188-190)

L181-2: "Colored network nodes" Statistical values could be included

Thank you for this suggestion. We added a reference to our paper Milke et al. (2022) where we analyzed how well interaction-adjusted index TINA (also based on SparCC network inference) describes ASV variation in the Pacific Ocean transect. This reflects how well co-occurrence structure correlates with environmental properties (l. 198-202).

L233-6: Were these modules identical or similar? to what percentage?

Thank you for this important question. The modules we identified in this study are solely based on the Atlantic and Pacific Ocean transect data. We assigned ASVs (whose identity is based on the pure 16S sequence and which is consistent among studies) to their modules and created a table that we uploaded in the supplementary material, showing the identity of ASVs, their 16S sequence and their modules. By classifying ASVs from other datasets based on this table, we could identify modules in the other datasets. This also means, that the modules we analyzed are all based on open-ocean samples. Hence, finding e.g. at SPOT that 60% of prokaryotic community abundance could be

associated to modules shows that 60% of all sequences were made up of open-ocean organisms, despite its proximity to the Californian coast. This gives an additional ecological dimension to the data that we now also mention in the discussion (l. 406-408).

L249: It is unclear what information is added by this analysis (SPOT). Is it just a validation?
See our answer above.

L273-5: Clarify
Done. (l. 314-317)

L329-31: This should be validated, with, for example, randomization analyses that show that the obtained modules differ from chance.
We agree but see answer above.

L347-9: Does this refer to richness or abundance? If referring to richness, it would be surprising that most diversity is participating in modules
We rephrased this sentence to make clear that we address abundance instead of richness (l. 411-414).

L351-2: Provide values for the lower proportions. Why should there be other modules? (additional taxa could be present and not be part of modules)

Thank you for this important point. We added lower proportions in the text (l. 414-418). As stated in the answer above, we did not infer modules for each dataset individually. Instead, we inferred module assignments from the two latitudinal transects. Co-occurrence networks present correlation-values between ASVs. This correlation can be hampered by the presence of too many ASVs due to p-value correction. Also, correlation values can be sensitive to the observed gradient: If you only analyze a small part of an (environmental) gradient, the correlation due to mutual environmental preferences will be hampered, because ASVs might be found in all samples. Hence, for the approach used here our heterogeneous sample set along the temperature gradient yielded optimal conditions to infer modules that reflect (sub-)communities that are differentially adapted to (temperature) conditions. However, as stated in the text, we can indeed miss certain modules that were not present in our sample set (e.g., because we only sampled below 20 m).

L355: "indicate that the oceanic prokaryotic communities exhibit a modular structure." Randomization analyses are needed to support this, in my opinion. That is, you probably need to show that random distributions of taxa do not lead to a modular structure.
See answer above. We added a comprehensive validation of modules.

L375-7: Unclear, clarify
Done. (l. 457-460)

L382-6: Unclear, clarify
Done. (l. 464-468)

L571: Indicate what edges mean. If using SparCC, edges may represent similar environmental preferences or ecological interactions. One way to assess whether edges could represent similar environmental preferences is using Flashweave, for example.
See answer above, we added a corresponding statement (l. 381-382, l. 388-391)

Reviewer #2 (Remarks to the Author):

In the submitted article, authors Milke et al examine V4-V5 16S rDNA amplicon sequences from 61 stations along two novel latitudinal ocean transects, from samples collected on the Malaspina expedition, and from the SPOT time series. They apply a correlation network analysis using the SparCC algorithm to identify co-occurring modules of taxa; use a combination of betaMNTD, the Raup-Crick index, and Bray-Curtis distances to estimate the relative importance of deterministic versus stochastic process on community assembly; and incorporate a large drifter dataset to estimate surface current transport matrices. The authors use genomic, bioinformatic, and statistical techniques

commensurate with the standards of the field. Additionally, their results demonstrating global prokaryotic community modules that are largely structured by homogenizing selection, as well specific regions with communities that are strongly influenced by current-driven dispersal, are particularly exciting.

However, I struggled to understand the environmental and taxonomic significance of the modules identified by the authors. The description of the modules in the text was muddled and the naming scheme (Table 1) did not add clarity. As a result, the biogeography of the modules presented in Figure 3 did convey any impactful result that I could easily interpret. I would implore the authors to more carefully craft their description of the modules and pair that description with a consistent color and naming scheme throughout the figures and text.

Thank you for this generally favorable comment on our study, your careful review and the constructive suggestions, which were a great help to further strengthen the manuscript. We want to point out, though, that these transects are not that novel as several papers already presented data of the Atlantic (Milici et al. 2016a, b, c, Milke et al. 2022, Front Microbiol, Dlugosch et al. 2022) and one paper the Pacific transect (Milke et al. 2022, ISME).

Thank you also for your comments on the modules. Combining many dimensions of information in a single figure is a challenging task and we tried to choose a color-theme that allows the reader to easily discern the different modules. Using e.g. temperature dependent colors on a red-blue-colormap makes it harder to differ between the modules. Also, there is not a single environmental variable such as temperature that explains the total variation. Instead, it would require at least two such dimensions (temperature AND depth) which would further complicate the color-coding and readability. We discussed this issue with several colleagues and agree that the Figure is very busy with three panels and thus difficult to read. It does contain very complex information and we felt unable to come up with a better visualization, considering the limited number of Figures allowed by this journal. We did change the naming scheme of the modules in Table 1 to be more consistent. We tried to use acronyms which intuitively relate to the upper and lower epipelagic, the major latitudinal regions and oceans. We further improved our description of the modules based on the comment below regarding niche-partitioning of the ecosystems.

Additionally, many of the arguments throughout the text and particularly in the Discussion were either worded poorly, had confusing structure, or were not well supported. I have endeavored to point out some of those instances in the minor comments below. I would encourage the authors to give this document a thorough examination to ensure that both the text and figures convey clear conclusions to the reader.

Thank you for your detailed remarks and comments which helped to smooth and focus the wording. We considered all points listed below and changed many parts throughout the manuscript.

In sum, this analysis shows promise to advance the fields of marine microbial ecology and biogeography. However, the conclusions conveyed by the text and figures lack clarity and, thus, it is difficult to interpret their overall significance.

We tried our best to consider your valuable comments and suggestions and hope that we succeeded to meet your criticisms in our attempt.

Minor Comments:

Lines 33-34: The structure of this sentence and the regions described is confusing. I am unsure whether additional commas are needed after "Indian Ocean" and/or "Mediterranean Sea."

We modified this sentence (l. 32-35).

Lines 39-40: What does concert the dispersal mean?

We changed the wording (l.39-40).

Line 78: these studies were missing or these studies missed

We changed the wording to make clear that these studies missed(l. 75-79).

Line 140: 50-70% of what?

We added the context of the percentage (l. 123).

Lines 146-147: Malaspina expedition, which was restricted to subtropical and tropical regions and a depth of 3 m, and which identified □

We changed the wording (l. 147-149).

Line 262: This seems like an odd place to finally name the modules. I would suggest including this information earlier when the modules are first being described.

We understand your argument and, in fact, thought also about introducing the description and names/acronyms of the modules earlier. However, we finally decided to fully describe the environmental context of their occurrence and introduce the names as an outcome of this description. When we introduced the names earlier, this context and the rationale for naming them would be missing. Therefore we would like to keep the naming at this place. We want to emphasize that we changed the acronyms (see Table 1) so that they become intuitively better understandable.

Line 271: relating □

Changed (l. 307).

Lines 281-284: Did the authors attempt to correlate tracer accumulation with modularity? According to the figure there are large areas with no overlap in estimates. However, it appears that the southern longitudinal transect has significant potential overlap between both metrics. Comparing the two metrics along this transect could be a powerful way to demonstrate a quantitative relationship between the authors two estimates of dispersal.

Thank you for this critique and suggestion. We made an additional analysis of the latitudinal transects where we measured tracer accumulation for each sampled station for various simulation times (Figure S15). We compared this metric against modularity and diversity of simultaneously present modules. We found that the heuristic measure of tracer concentration works best along the latitudinal transects. Tracer concentration is an artificial approximation of aggregation zones which is, e.g., used in microplastic studies as well, but its units of measure are highly artificial and rather give an approximation of regions where currents would lead to convergent and divergent zones. In our case it was most suitable to use the latitudinal data to highlight the overlapping between aggregation zones and modularity. Along the longitudinal transect, borders of aggregation zones were less distinct and also strongly depending on simulation times. We made an additional statement in the discussion where we highlight the fact that tracer concentrations yield only approximate regions of particle aggregation that requires in depth analyses in future studies (l. 557-559). Further, we also made an additional analysis of the average current speed based on the global drifter dataset that clearly shows how aggregation zones are linked to regions of minimal average currents (l. 320-326).

Lines 308-310: The language here is difficult to understand. Also, I believe there is a “without a corresponding i) □

We modified this sentence (l. 358-361)

Line 315: also include □

Changed (l. 366).

Lines 317-320: This claim is not well supported, there has been significant effort to characterize prokaryotic biogeography in the surface ocean.

Thank you for this comment. We clarified this statement, as it was not focussing on current induced dispersal. We are aware of the comprehensive work of scientists such as the Mick Follows group (MIT) who did a lot of analyses on current induced distribution of phytoplankton, but want to highlight here that we are exclusively looking at prokaryotes, even though of different size fractions. Our biogeography, in particular of the FL prokaryotes, strongly differs from larger microbes.

Line 320: It has been applied □ What are the authors referring to here?

We clarified this statement (l. 371-374).

Lines 322-325: This is a run-on sentence. Please rephrase.
Changed (l.374-377).

Line 329: Is referring to co-occurrence cluster analysis?
We now clarify this with another sentence (l. 381-382)

Lines 353-355: This is circular reasoning. The authors used a statistical technique designed to identify co-correlated clusters of taxa, i.e. modules. Thus, is it unsurprising that "oceanic prokaryotic communities exhibit a modular structure."

We understand your reasoning, However, feel that this is not entirely true: The modularity of a network is a continuous variable that can be low (no modular structure) or high (strong modular structure with discrete clusters). We identified here that the co-occurrence network of prokaryotes along a latitudinal gradient shows very high modularity that led to discrete modules. Therefore, we want to keep this statement.

Lines 375-379: How would the authors identify the source region where a theoretical bacterial species originates from? Could the co-occurrence of modules simply imply multidimensional niche partitioning of the biotope in the classical Hutchinsonian sense (see for instance Colwell and Rangel 2009)?

Thank you for bringing up this important point. This prompted us to make further analyses regarding the niche space and analyzed niche-spaces of ASVs and modules based on a CCA using environmental parameters that were found to significantly influence ASV variation (Figs. S9 & S10). Parameters were chosen based on forward-selection model building. We could find very similar positions in the realized niche-space for ASVs and their respective modules. ASVs of the same module also grouped in the CCA biplot, revealing that modules reflect similar environmental niches. However, applying the CCA to modules yielded >3x higher inertia explained. We present this new and most valuable analysis in l. 252-264 and added a respective paragraph in the Methods.

Line 404: do not constitute more than 20%, and usually less than 10%, of the total
Changed (l. 486-487).

Line 406: Why are ecological mechanisms more important for free living prokaryotes? Is drift not an ecological mechanism?
You are right, we changed that to "deterministic processes" (l. 488)

Lines 454-456: This sentence structure is confusing, please rephrase
Changed (l. 550-552).

Line 483: delete well
Changed. (l. 581-582)

Lines 481-485: Run-on sentence. Please split into two sentences.
Done. (l. 580-584)

Line 510: It is encouraging that 90% of the ASVs were observed in both ocean basins (Lines 116-124), but the differences in extraction technique between the Atlantic and Pacific samples could contribute to the 10% differences observed between the two basins. The authors should outline what differences exist between the two extraction kits. Alternatively, were any replicate samples extracted with each technique and compared?

We used the same extraction kit. The company MoBio was taken over by Qiagen a couple of years ago, but they used the exact same chemistry in the PowerSoil kit that we used. They recently changed the chemistry of the lysate, resulting in the PowerSoil Pro kit, but that happened after we extracted the respective samples. To make that clear we added a statement in the methods section (l. 609-610).

Line 514: Illumina
Changed. (l.613)

Table 1: I find the names chosen confusing, as they do not appear to use a systematic structure. For instance, "lower epipelagic" is referred to with both an "L" (as in PAL-C) and with an "LE" (as in ULE-C). Similarly, the word "temperate" is referred to with "TEM" (TEM-C), "T" (TSP-C), and "TR" (TRP-C). Or, as another example, I am not sure what the difference is between "lower epipelagic" and "Low-Light."

We agree and changed the naming scheme of the modules to be more consistent and intuitively understandable for what they stand for. See comment above.

Figure 1: I would encourage the authors to reconsider the layout of this figure. As currently configured it is hard to follow. Perhaps having the richness plots aligned latitudinally with their associated ocean basin map (with the proportion plots underneath) would assist?

We tried to relocate the richness plot and align it to the maps, but this made the figure worse in our opinion. Therefore, we stick to the layout of the figure.

Figure 2: In (b) on the figure move the Temperature up to a single line

Thank you for pointing that out. Something went wrong in the conversion to PDF and we fixed that issue.

Figure 3: I find it hard to interpret what the changes in the relative abundance of the modules mean from a taxonomic, environmental, or functional perspective. The authors might consider a color scheme denoting the module's environmental association (i.e., shades of red color for "Modules 2, 4, 8 and 9," shades of another color for "Modules 3, 6 and 10," and a final color for Modules 1, 5 and 7).

As noted in the answer above, we chose the color scheme as it helped to easily discern the various modules in the figures. Therefore, we want to stick to the colors, but improved the description of the modules in the text (see niche-space analysis, renaming of modules and various changes in the manuscript).

Reviewer #3 (Remarks to the Author):

This manuscript uses two longitudinal transects in the Pacific and Atlantic oceans to construct prokaryotic species modules based on co-occurrence of ribosomal DNA variable region genes. The authors found sometimes similar modules in both oceans and relate their distributions with abiotic and biotic factors. Finally, they study the roles of selection and biased dispersal in their distributions, suggesting a strong role for current systems in module distributions.

The use of new data sets along latitudinal transects is of interest, as well as the characterization of conserved modules. Their validation by external data, and some interpretations based on transport by currents are also of interest. However, the paper does not completely integrate this study in the current literature. For instance, it should be discussed with the results of Chaffron et al. (Science Adv 2021) regarding the modular structure of plankton communities, or with Sommeria-Klein et al. (Science 2021) for the effects of dispersion according to taxonomic groups and biogeography (for eukaryotes).

Thank you for your favorable review and comments and constructive suggestions and pointing that out the missing literature context. We added a discussion of the mentioned literature in the text l. 389 and 400-401.

It looks rather superficially to the influence of currents. The water masses are not described according to their temperature/salinity characteristics, which are important for advection studies.

In our analyses we used a Lagrangian approach to identify regions of particle accumulations that we assume also leads to mixing of different microbial modules. In these regions, water masses would mix and strong ocean fronts form sharp gradients of temperature and salinity which hamper the classification of discrete surface water masses. Therefore, we focused our analyses on tracer

concentration values derived from the global drifter dataset rather than the analysis of water masses. Nonetheless, in a previous publication from Giebel et al. (2021) a detailed description of the water masses in the Pacific Ocean transect was made and we added a corresponding reference (l.117-120).

It is to be discussed if latitudinal sampling is the best for studying current effects. The simulation with drifters may help for this, but it is unclear how the authors use it and reach their advection time of 2-3 years. More details and figures appear necessary for this analysis.

Relating to the answer to a comment of Reviewer 2, we highlighted that latitudinal sampling works best for this heuristic method to infer aggregation zones. We added a corresponding statement in the discussion (l. 326-333). Further, we added an additional analysis of the current system built on the global drifter dataset to display average surface currents (Fig. S14). This analysis shows that latitudinal sampling highlights the dynamics in the current gradients, whereas longitudinal sampling will not capture the same current dynamics. We also made an additional paragraph in the methods section where we explain the construction of the transport matrices in more detail (l. 766-773). Simulation times were chosen such that tracer concentrations display approximate accumulation regions formed by large-scale transport.

More generally, some results do not seem to be based on clear statistical methods, that should be better explained and their limitations indicated.

As stated in the answer to reviewer 2, we highlighted in the discussion that tracer concentrations are an artificial measure of aggregation zones derived from drifter data, that yields only approximate regions where particles would accumulate. This prevents statistical robust analyses such as correlations. Instead, we made an additional analysis to demonstrate how well aggregation zones and modularity overlap close to the subtropical fronts (Fig. S14).

The sequencing effort per sample is apparently not indicated. As the number of recovered ASVs seems relatively low, this should be clarified. What part of the diversity is accessed with this sequencing depth, and how does it impact modules?

Thank you for pointing that out. We added a statement in the Results where we show sequencing depth per sample, but also highlight the original references of the transect data, where in depth analyses of the nucleotide data is presented (l. 88-91). We want to highlight, though, that we used a strict abundance filter as stated in the text. This abundance-filter removes low abundant ASVs from the data that would prevent robust co-occurrence inference. Similar approaches have been done by other studies before that did co-occurrence analyses (e.g. also in Chaffron et al. 2021). A more detailed analysis of the applied abundance filter and its impact on biogeographic analyses was done in another publication (Milke et al. 2022: *Frontiers in Microbiology*).

The analyses on ASVs seem to be done with relative abundances, do the authors also test the robustness of their conclusions using rarefactions?

As stated in the original SparCC publication of Friedman and Alm (2012) we did not rarefy the count data prior to network inference. Rarefaction is not necessary due to the data transformation that is involved in the SparCC algorithm. We rarefied our data prior to all alpha-diversity measures (richness and Effective Number of Species ENS) as described in the methods section (l. 723-725).

Lines 114-135, it may be interesting to have more information about the taxa that are unique or common between basins. It is unclear if the results of the richness analysis along the latitudinal gradient is new compared to Ibarbalz et al.?

Thank you for this suggestion. However, we feel that we basically cover this point. We refer to our analysis of the basin-unique taxa in Figure S2, where we show which taxa are unique in each ocean basin and where they are abundant. The corresponding text in the Results is in l. 122-132. Further, we added a comparison with the results of Ibarbalz et al. (2019) where we compare the latitudinal diversity gradients (l. 541-544).

Line 233, the comparison with Malaspina data is of interest, can the authors discuss the potential effects of different sampling/sequencing methods and depths?

We added a statement where we highlight the differences to the Malaspina data in l. 492-502.

Line 255 and Fig5A, the correlation between module abundances and El Niño is not obvious, and may be due to random effects. A statistical analysis is necessary.

We tested whether abundance ratio between warm- and cold-water modules 2 & 6 was significantly different during El-Niño as compared to the other time frames and found significant differences that we further mention in the text (l. 287-292). As the El-Niño event was a single event within the sampled timeframe, we are unaware how to test that this effect did not result from random variation. However, a recent publication from Yeh et al. (2023, ref. 28) made an in-depth analysis of the warming effect induced by El-Niño on the timeseries, hence further supporting our assumption and validating the ecological significance of our modules. We added the corresponding reference in the text to the Discussion (l. 403-406).

Line 225, it is well known that biogeography is more pronounced in surface than in deep samples, the literature should be better cited here.

Thanks for pointing that out, we added a reference here (l. 242-245).

Fig5B, it is interesting to see that the time series does not show large seasonal variations in module abundances. This seems to reinforce the conclusions, and may be further described.

Indeed, this was not well discussed before. We highlighted in the discussion how the temporal stability matches to the marginal seasonal variation of the large-scale current system in the observed region (l. 406-408).

REVIEWERS' COMMENTS

Reviewer #1 (Remarks to the Author):

The authors have satisfactorily addressed all my previous comments. I have no further comments.

Reviewer #3 (Remarks to the Author):

The authors responded adequately to most of my concerns by adding discussions and references that clarify the impact of the results. One minor point that may need additional comments is the way sequencing depth and filters for abundance are applied, which limits the study to abundant organisms. The authors point that this is done in other studies to provide robust modules, but it certainly affects the conclusion that a large part of the modules is shared between basins and oceans. The large majority of ASVs is not considered using this filtering. This is mentioned, but a specific sentence may be added to highlight this point.